# Continual Learning via Learning Continual Memory in Vision Transformer

## Abstract

This paper explores continual learning (CL) using Vision Transformer (ViT) in streaming tasks under the challenging exemplar-free class-incremental (ExfCCL) setting. We formulate ExfCCL as a learning problem consisting of two key sub-systems: (i) task ID inference for test data, which selects appropriate task-specific head classifiers to accounting for varying class distributions across tasks and streams, and (ii) a dynamic learning-to-grow feature backbone that balances stability and plasticity, mitigating catastrophic forgetting through task synergies. Following the common protocol that the first task can train a ViT sufficiently well as the base model, we address these sub-systems from a continual memory learning perspective. To support task ID inference, we utilize an external memory mechanism that maintains task centroids computed by the base ViT throughout CL. For the feature backbone, we identify optimal placements for internal (parameter) memory to enable a dynamic, task-synergy guided growing feature backbone. We propose a Hierarchical Exploration-Exploitation (HEE) sampling-based neural architecture search (NAS) method that effectively learns task synergies by continually and structurally updating internal memory with four basic operations: `reuse`, `adapt`, `new`, and `skip`. Our approach, dubbed **Continual Hierarchical-Exploration-Exploitation Memory (CHEEM)**, is evaluated on the challenging Visual Domain Decathlon (VDD) and ImageNet-R benchmarks, demonstrating its effectiveness.

## 1 Introduction

Developing continual learning machines is a key objective in Artificial Intelligence (AI), aiming to replicate human-like adaptability and the ability to learn-to-learn, enabling proficiency in streaming tasks. Despite their advances, state-of-the-art Deep Neural Networks (DNNs) still lack true biological intelligence in the realm of continual learning and are particularly hindered by the critical issue of *catastrophic forgetting* when exposed to streaming tasks in dynamic environments (McCloskey & Cohen, 1989; Thrun & Mitchell, 1995).

To address catastrophic forgetting, two primary categories of continual learning methods have emerged: exemplar-based methods (Aljundi et al., 2019b; Hayes et al., 2019; Wu et al., 2019) and exemplar-free methods (Kirkpatrick et al., 2017; Li et al., 2019; Wang et al., 2022d;c;a). While both have shown promising progress, exemplar-free methods are especially appealing due to their ability to learn new tasks without retaining any data from previous tasks. Furthermore, continual learning has evolved from the traditional task-incremental protocol, where task IDs of test data are available during inference, to the more challenging class-incremental protocol, in

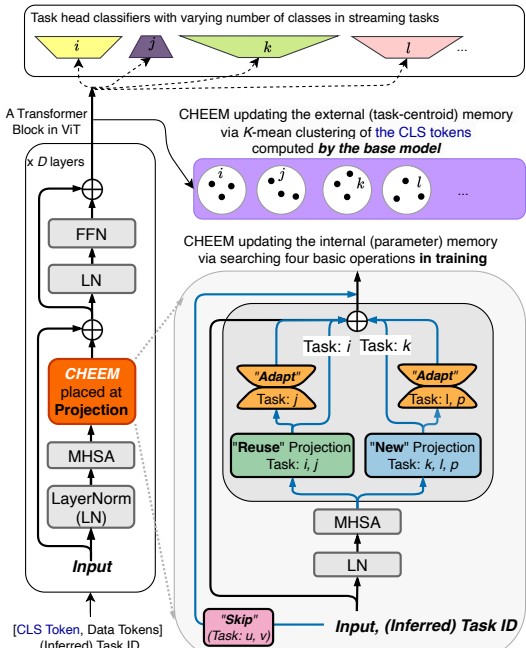

Figure 1: Illustration of the proposed CHEEM.

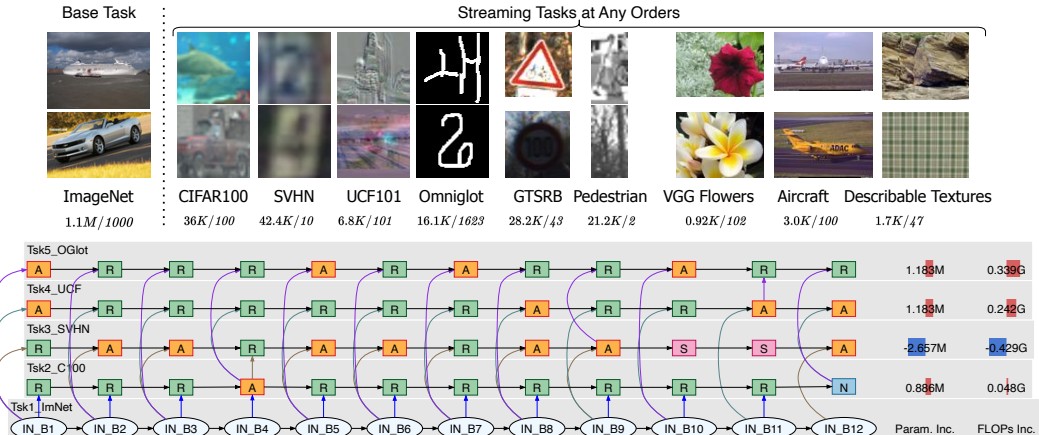

Figure 2: **Top:** The challenging VDD benchmark (Rebuffi et al., 2017a) consisting of 10 tasks of different nature with `#training images/#classes` significantly varying across different tasks. **Bottom:** Starting from the base task (Tsk1_ImNet) trained ViT (Dosovitskiy et al., 2021) (e.g., the 12-layer ViT-B model consisting of IN_B1 to IN_B12), together with the best performance on VDD in experiments, our CHEEM can learn sensible memory structures: *e.g.*, Tsk3_SVHN learns to `skip` both B10 and B11 with both mode complexity and FLOPs reduced. Tsk4_UCF and Tsk5_OGlot learn to `adapt` the first layers to account for the domain shifts. We show the first 5 tasks for clarity (see more in the supplementary). The last two columns show the number of new task-specific parameters and added FLOPs, in comparison with the base model, subject to R euse, A dapt, N ew and S kip.

which task IDs are unknown during inference. This paper focuses on exemplar-free class-incremental continual learning (ExfCCL).

Recently, the emergence of powerful pretrained Transformer models (Vaswani et al., 2017; Dosovitskiy et al., 2021; Radford et al., 2021) has driven significant interest in ExfCCL using pretrained and frozen Vision Transformers (ViTs) (Dosovitskiy et al., 2021), primarily explored through the lens of prompt-tuning or prefix-tuning (Wang et al., 2022d;c;a; Smith et al., 2023a). However, this approach presents two main drawbacks: (i) A single frozen pretrained Transformer backbone cannot accommodate all streaming tasks of diverse nature, such as those found in the VDD benchmark (Rebuffi et al., 2017a), shown in Fig. 2. While this method maximizes the stability of the feature backbone, it relies on input or prefix prompts to address plasticity by explicitly leveraging the self-attention mechanisms. (ii) Due to the quadratic complexity of ViTs concerning the number of input tokens, incorporating task-specific input prompts or layer-wise prefix prompts into a frozen pretrained Transformer backbone significantly increases computational demands. Continual learning where computational requirements always grow for new tasks by design regardless of task complexity fails to reflect true intelligence.

The challenge of structurally and dynamically updating a pretrained Transformer backbone for ExfCCL remains an open problem, as it requires achieving a balance between the backbone's stability and plasticity while enabling dynamic computation and mitigating catastrophic forgetting.

Moreover, ExfCCL involves tackling the challenge of designing task head classifiers, which can either explicitly infer task IDs to select task-specific head classifiers (Wang et al., 2022a) or share a growing, common head classifier (Wang et al., 2022d;c; Smith et al., 2023a). The former approach is straightforward and can help in learning task-specific components and enhancing the plasticity of the backbone, but may suffer from low average precision of the task ID inference for certain tasks. In contrast, the latter approach resorts to more sophisticated prefix prompt tuning to induce plasticity in the feature backbone and can create a discrepancy between training, where head segments from previous tasks are masked out in the softmax function of loss computation, and testing, where the entire head is used to infer class labels. This discrepancy can lead to instability when dealing with streaming tasks that involve a varying number of classes among tasks, such as those encountered in the VDD benchmark.

**Our aim** in this paper is to study ExfCCL using ViTs. As illustrated in Fig. 1, we formulate it as a problem of learning continual memory in ViT, which has two components: (i) The internal

memory enabling a dynamic learning-to-grow feature backbone that balances stability and plasticity, mitigating catastrophic forgetting through **task synergies, in which a new task learns automatically to reuse/adapt modules from previous similar tasks, or to introduce new modules when needed, or to skip some modules when it appears to be an easier task** (see the bottom of 2). The internal parameter memory learning presents alternative perspectives to the input and prefix prompting based methods (Wang et al., 2022d;c; Smith et al., 2023a; Wang et al., 2022a). (ii) The external memory enabling task ID inference for test data, for which we adopt a method proposed in (Wang et al., 2022a) for its simplicity.

We follow the common protocol in the prior art that the first task (e.g., ImageNet-1k (Russakovsky et al., 2015)) can train a ViT sufficiently well as the base model. To enable a dynamic, learning-to-grow feature backbone starting from the base model, we identify and provide simple yet effective solutions for two key challenges: **(i) Which modules in a ViT should be reused, adapted, newly created, or skipped for a new task in ExfCCL?** It is computationally impractical, and counterproductive to the stability-plasticity trade-off, to make all ViT components dynamic (as opposed to a fully frozen model). We designate the output projection layer following multi-head self-attention (MHSA) as *the task-synergy internal (parameter) memory* that will be structurally and dynamically updated. **(ii) How can we represent and continually learn this task-synergy internal memory to enable dynamic memory structures across streaming tasks?** We propose organizing the memory using a mixture of experts (MoEs) similar in spirit to (Riquelme et al., 2021), as depicted in Fig. 1 and illustrated through the "task-factorized" visualization in Fig. 2. To learn to select the optimal synergy operation from the four choices (`reuse`, `adapt`, `new`, and `skip`), we introduce an effective hierarchical exploration-exploitation (HEE) sampling-based neural architecture search (NAS) method. This approach is inspired by the task-incremental learn-to-grow method (Li et al., 2019). Our proposed method is termed **CHEEM** (*Continual Hierarchical-Exploration-Exploitation Memory*).

**Our Contributions**. This paper makes three main contributions to the field of exemplar-free class-incremental continual learning (ExfCCL) using ViT. (i) It presents a hierarchical task-synergy exploration-exploitation sampling based NAS method for learning task-aware dynamic models continually with respect to four operations: `Skip`, `Reuse`, `Adapt`, and `New` to mitigate catastrophic forgetting. (ii) It identifies a "sweet spot" in ViT as the task-synergy internal (parameter) memory, i.e., the output projection layers after MHSA in ViT. It also presents a new usage for the class-token `CLS` in ViT as the internal memory updating guidance, in addition to leveraging it in maintaining the external (task-centroid) memory for task ID inference on the fly. (iii) It is the first work, to the best of our knowledge, to evaluate continual learning with ViTs on the large-scale, diverse and imbalanced VDD benchmark (Rebuffi et al., 2017a), with better performance than the prior art.

## 2 APPROACH

### 2.1 PROBLEM FORMULATION OF CHEEM IN EXFCCL

We start with a vanilla $D$-layer ViT model (e.g., the 12-layer ViT-Base) (Dosovitskiy et al., 2021). The left of Fig. 1 shows a ViT block. Denote by $x_{L,d}$ an input sequence consisting of $L$ tokens encoded in a $d$-dimensional space. In ViTs, the first token is the so-called class-token, `CLS`. The remaining $L-1$ tokens are formed by patchifying an input image and then embedding patches, together with additive positional encoding. A ViT block is defined by,

$$z_{L,d} = x_{L,d} + \text{Proj}(\text{MHSA}(\text{LN}_1(x_{L,d}))), \qquad (1)$$

$$y_{L,d} = z_{L,d} + \text{FFN}(\text{LN}_2(z_{L,d})))), \qquad (2)$$

where $\text{LN}(\cdot)$ represents the layer normalization (Ba et al., 2016), and $\text{Proj}(\cdot)$ is a linear transformation fusing the multi-head outputs from MHSA module. The FFN is often implemented by a multi-layer perceptron (MLP) with a feature expansion layer $\text{MLP}^u$ and a feature reduction layer $\text{MLP}^d$ with a nonlinear activation function (such as the GELU (Hendrycks & Gimpel, 2016)) in the between.

Denote by $\mathcal{T} = \{T_1, \cdots, T_t, \cdots, T_N\}$ a stream of tasks in continual learning, where each task $T_t$ consists of a training set $D_t^{train}$ and a testing set $D_t^{test}$. We make no restrictive assumptions regarding the nature, order, or number of classes in streaming tasks, either per task or in total. For example, there are 9 diverse, streaming tasks in the VDD benchmark (2).

Denote by $(f_1, C_1)$ the ViT (Dosovitskiy et al., 2021) base model (e.g., the 12-layer ViT-Base) trained on the first task $T_1$ (e.g., ImageNet-1k (Russakovsky et al., 2015)), where $f_1$ is the backbone and $C_1$ the head classifier. Denote by $(\mathcal{F}_t, \mathcal{C}_t)$ the sequentially and continually learned model after task $T_t$

(for $t \geq 1$). $\mathcal{F}_t$ is the super ViT backbone structurally and dynamically updated from the base model $f_1 = \mathcal{F}_1$, and $\mathcal{C}_t = \{C_1, \cdots, C_t\}$ consisting of task-specific head classifiers each of which is trained from scratch. Let $\Theta_t = \mathcal{F}_t \setminus \mathcal{F}_{t-1}$ be task-specific backbone parameters, which we term **the internal parameter memory** in ExfCCL to exploit task synergies. We note that ExfCCL requires no use of exemplar data of previous tasks in any forms in learning $(\Theta_t, C_t)$ for task $T_t$.

In inference, since the task IDs of test data are unknown. For a test sample $x$, we will need to infer its task ID on the fly. Following the prior art (Wang et al., 2022d;c; Smith et al., 2023a; Wang et al., 2022a), we use the base model $f_1(\cdot)$ as the query function $q()$, and use $q(x)$ to retrieve the task ID from **the external memory** for which we adopt the method proposed in (Wang et al., 2022a) for its simplicity. With task ID available (provided in training or inferred in testing), we can allocate the task-specific backbone $f_t \subset \mathcal{F}_N$ for task $T_t$. The task-specific model is then specified by $(f_t, C_t)$.

## 2.2 CONSTRUCTING THE EXTERNAL MEMORY

We choose to explicitly infer task IDs for test data (see Appendix A for the analyses). We adopt the method proposed in S-Prompts (Wang et al., 2022a). For a task $t$, we leverage $K$-mean clustering of the CLS tokens of its training images computed by the query function $q(\cdot)$ (i.e., the base model $f_0$). Denote by $Z_t = \{z_t^1, \cdots, z_t^K\}$ the clustered task centroids for task $T_t$, where for simplicity we use the same $K = 5$ across tasks. We have the external memory, $\Psi = \cup_{t=1}^N Z_t$ after training. In inference, for a testing sample $x$, we first compute its CLS token using the same query function $q(x)$, denoted by $z$. The task ID is then inferred via $K$-NN retrieval, $KNN(z, \Psi)$, either by retrieving the class ID of the Top-1 NN centroid in the external memory $\Psi$, or by majority voting of the class ID from the $K$-NN centroids.

We note that the such constructed external memory does not break the exemplar-free protocol since we do not use exemplars in any forms from previous tasks. We also note that we focus on the learning of the internal parameter memory. The decoupled design between the external memory and the internal memory will enable us to integrate and/or develop more advanced task ID inference approaches in future work, while potentially promoting our proposed CHEEM in the field of ExfCCL.

## 2.3 IDENTIFYING THE TASK-SYNERGY INTERNAL MEMORY

The proposed identification process is straightforward. Without introducing any modules handling forgetting, we compare both the task-to-task forward transferrability and the sequential forgetting for different components in a ViT block. **Our intuition is that a desirable component for placing the task-synergy parameter memory must enable strong transferrability with manageable forgetting, while being lightweight to account for the trade-off between stability and plasticity.**

To that end, we use the VDD benchmark (Rebuffi et al., 2017a) (see Fig. 2). We first train a ViT-Base (Dosovitskiy et al., 2021) on the first task, ImageNet (Russakovsky et al., 2015), as the base model $f_1(\cdot)$. To measure the task-to-task transferability, we *individually fine-tune* $f_1$ in a task-to-task transfer learning manner for the remaining 9 streaming tasks. Let $f_{t|1}$ be the backbone fine-tuned for task $T_t$ (for $t \geq 1$), and $C_t$ the head classifier trained from scratch. The average Top-1 accuracy is:

$$AA = \frac{1}{N} \sum_{t=1}^{N} \text{Acc}(T_t; f_{t|1}, C_t) \qquad (3)$$

where $\text{Acc}()$ uses the Top-1 classification accuracy.

Table 1: Results of identifying the optimal placement of our CHEEM in ViT by testing 11 components (Eqns. 1 and 2).

| Index | Finetuned Component | $A\mathbf{A}$ (Eqn. 3) | $A\mathbb{F}$ (Eqn. 4) |
|---|---|---|---|
| 1 | $LN_1 + LN_2$ | 81.76 | 21.24 |
| 2 | FFN | 84.20 | 44.76 |
| 3 | $MLP^d$ | 83.66 | 37.99 |
| 4 | $LN_2$ | 80.04 | 16.35 |
| 5 | $MHSA + LN_1$ | 85.26 | 54.38 |
| 6 | $LN_1$ | 81.18 | 19.04 |
| 7 | Query | 81.57 | 19.69 |
| 8 | Key | 81.56 | 19.19 |
| 9 | Query+Key | 81.49 | 31.10 |
| 10 | Value | 84.99 | 37.58 |
| 11 | Projection (**CHEEM**) | 85.11 | 30.50 |
| | Classifier w/ Frozen Backbone | 70.78 | - |

To measure the sequential forgetting, we *continually fine-tune* the backbone started from $f_1$ on the 9 tasks in a randomly sampled and fixed streaming order (as shown in Fig. 2). Let $f_{1:t}$ be the backbone trained sequentially and continually after task $T_t$ and $C_t$ is its head classifier. The average forgetting (Chaudhry et al., 2018) on the first $N - 1$ streaming tasks is,

$$A\mathbb{F} = \frac{1}{N-1} \sum_{t=1}^{N-1} \left( \max_{j \in [t,N]} a_{j,t} - a_{N,t} \right), \qquad (4)$$

where $a_{j,t} = \text{Acc}(T_t; f_{1:j}, C_t)$.

As shown in Table 1, we compare 11 components or composite components in ViT. Consider the strong forward transfer ability, manageable forgetting, maintaining simplicity and for less invasive implementation in practice, **we select the Projection layer after the MHSA as the task-synergy internal (parameter) memory** (Fig. 1) to realize our proposed CHEEM for ExfCCL. We note that the last row of Table 1 shows the result of a conventional transfer learning setting in which only the head classifier is trained with the backbone frozen, which clearly shows that one frozen backbone can not fit all streaming tasks, as aforementioned and entailing dynamic updating of the backbone in continual learning.

## 2.4 LEARNING THE TASK-SYNERGY INTERNAL MEMORY

The proposed internal memory of our CHEEM is represented by a MoEs. Starting with the base ViT model $f_1$, the internal memory at the $l$-th layer in ViT consists of a single expert defined by a tuple,

$$\mathrm{E}_l^{(1,)} = (\theta_l^{(1,)}, \mu_l^1), \tag{5}$$

where the subscript represents the layer index and the list-based superscript shows which task(s) use this expert. $\theta_l^{(1,)}$ are the parameters of the projection layer and $\mu_l^1 \in R^d$ is the associated mean class-token (CLS) pooled from the training dataset after the model is trained, which is task specific (as indicated by the superscript). For example, if an expert is reused by another task (say, 3) in continual learning, we will have $\mathrm{E}_l^{(1,3,)} = (\theta_l^{(1,3,)}, \mu_l^1, \mu_l^3)$.

As shown in Fig. 3, for a new task $t$, **learning to update CHEEM consists of:** i) the Supernet construction, ii) the Supernet training, and iii) the target network selection and finetuning.

### 2.4.1 SUPERNET CONSTRUCTION VIA `Reuse`, `Adapt`, `New` AND `Skip`

For clarity, we consider a single layer $l$ for a new task, and the current memory consists of two experts, $\{\mathrm{E}_l^{(1,)}, \mathrm{E}_l^{(2,)}\}$ (Fig. 3, left). We utilize four operations in the Supernet construction:

- R euse the projection layer from a previous (similar) task for a new task.
- A dapt by adding a new lightweight layer on top of the projection layer of a previous task, implemented by a MLP with one squeezing hidden layer.
- N ew by adding a new projection layer, which enables the model to handle dissimilar situations.
- S kip the entire MHSA block, which encourages the adaptivity accounting for the diverse nature of tasks.

The Supernet is constructed by `reusing` and `adapting` each existing expert at layer $l$, and adding a `new` and a `skip` expert (the bottom of Fig. 3). The newly added `adapt` MLPs and `new` projection layers will be trained from scratch using the data of a new task only. The right-top of Fig. 3 shows the `Adapt` operation on top of $\mathrm{E}_l^{(2,)}$ is learned and added, $\mathrm{E}_l^{(3,)} = (\theta_l^{(3,)}, \mu_l^3)$ where $\theta_l^{(3,)}$ represents parameters of the `adapt` MLPs learned for the task 3, and $\mu_l^3$ is the mean CLS token pooled for the task 3. The expert $\mathrm{E}_l^{(2,)}$ is updated to $\mathrm{E}_l^{(2,3,)} = (\theta_l^{(2,3,)}, \mu_l^2)$ indicating its weights are shared with task 3. We provide implementation details for `Adapt` in the Appendix D.5.

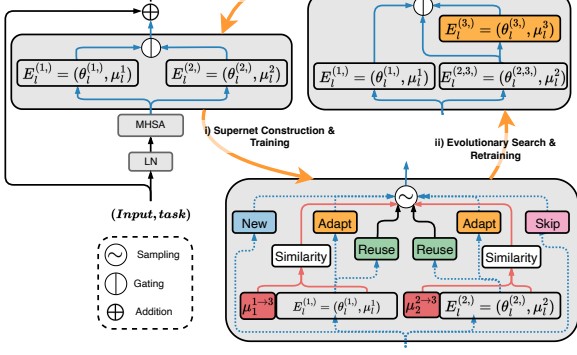

Figure 3: Illustration of CHEEM learning via NAS.

### 2.4.2 SUPERNET TRAINING VIA THE PROPOSED HEE SAMPLING-BASED NAS

To train the Supernet constructed for a new task $t$, we build on the SPOS method (Guo et al., 2020) due to its efficiency. The basic idea of SPOS is to train a single-path sub-network from the Supernet by sampling an expert at every layer in each mini-batch of training. One key aspect is the sampling strategy. The vanilla SPOS method uses uniform sampling (i.e., the *pure exploration* (PE) strategy, Fig. 4 left). **We propose an exploitation strategy** (Fig. 4 right), which utilizes a hierarchical sampling

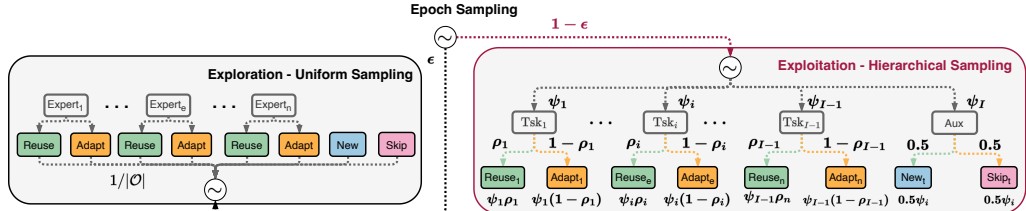

Figure 4: Illustration of the proposed hierarchical task-synergy exploration-exploitation (HEE) sampling based NAS. It integrates the vanilla exploration strategy (left) and the proposed exploitation strategy (right) with an epoch-wise scheduling.

method that forms the categorical distribution over the operations in the search space *explicitly based on task synergies computed based on the pooled task-specific* CLS *tokens*.

As illustrated in the right of Fig. 4, at each layer $l$ in the Supernet, for a new task $t$, our proposed 3-level HEE sampling is realized by:

- **The epoch-level sampling:** At the beginning of an epoch in the Supernet training, we choose the pure exploration strategy with a probability of $\epsilon_1$ (e.g., 0.3), and the exploitation strategy with a probability of $1 - \epsilon_1$.
- **The task-level sampling:** There are $t - 1$ previous tasks, some of which may use Skip at the $l$-th layer and will not be used in sampling. We introduce an auxiliary task to handle sampling New and Skip for the current task $t$. *The task-level sampling is based on a categorical distribution* $(\psi_1, \cdots, \psi_i, \cdots, \psi_{I-1}, \psi_I)$, which is computed by the Softmax function over the similarity scores defined below, where $I \leq t$.
- **The operation-level sampling:** With a previous task $i$ sampled with the probability $\psi_i$, we sample the Reuse and Adapt operation using a Bernoulli distribution $(\rho_i, 1 - \rho_i)$, where $\rho_i$ is the success rate computed by the Sigmoid function of the task similarity score (defined below).

**The Task Similarity Score.** Consider one expert $\mathbb{E}_l^{(i,j,)}$ at the $l$-th layer which is shared by two previous tasks $i$ and $j$ with their mean CLS tokens $\mu_l^i$ and $\mu_l^j$ respectively, we compute the mean CLS tokens using the models for task $i$ and $j$ on the training dataset $D_t^{train}$ of the current task $t$, $\mu_l^{i \to t}$ and $\mu_l^{j \to t}$. The task similarity between task $i$ and $t$ is computed by,

$$S_l^{i,t} = \texttt{NormCosine}(\mu_l^i, \mu_l^{i \to t}), \tag{6}$$

where $\texttt{NormCosine}(\cdot, \cdot)$ is the Normalized Cosine Similarity, which is calculated by scaling the Cosine Similarity score between $-1$ and $1$ using the minimum and the maximum Cosine Similarity scores from all the experts in all the MHSA blocks of the ViT. This normalization is necessary to increase the difference in magnitudes of the similarities between tasks, which results in better Expert sampling distributions during the sampling process in our experiments.

**The Auxiliary Task.** For a new task $t$, to handle the New and Skip experts at each layer $l$ for which we do not have direct similarity scores. Instead, we introduce an auxiliary task, Aux (see the right of Fig. 4) which gives equally-likely chance to select New or Skip expert. For the Aux task itself, the similarity score between it and task $t$ is defined by,

$$S_l^{aux,t} = -\max_{i=1}^{t-1} S_l^{i,t}, \tag{7}$$

which intuitively means we probabilistically resort to the New operation or the Skip operation when other experts turn out not "helpful" for the task $t$.

### 2.4.3 TARGET NETWORK SELECTION AND FINETUNING

After the Supernet is trained, we adopt the same evolutionary search used in the SPOS method (Guo et al., 2020) based on the proposed HEE sampling strategy with a different epoch sampling probability $\epsilon_2$ ($\epsilon_2 > \epsilon_1$, e.g., $\epsilon_2 = 0.5$, to encourage more exploration during the evolutionary search). The evolutionary search is performed on the validation set to select the path which gives the best validation accuracy. After the target network for a new task is selected, we retrain the newly added layers by the New and Adapt operations from scratch (random initialization), rather than keeping the weights from the Supernet training. This is based on the observations in network pruning that it is the neural architecture topology that matters and that the warm-up weights may not need to be preserved to ensure good performance on the target dataset (Liu et al., 2019b).

Table 2: Results on the VDD benchmark (Rebuffi et al., 2017a) using ViT-B/8 (Dosovitskiy et al., 2021) over 3 different seeds. The last two columns show the performance under the task-incremental continual learning (TCL) setting.

| Method | C100 36k/100 | SVHN 42.4k/10 | UCF 6.8k/101 | OGlt 16.1k/1623 | GTSR 28.2k/43 | DPed 21.2k/2 | Flwr 0.92k/102 | Airc. 3.0k/100 | DTD 1.7k/47 | $A\mathbb{A}$ (Eqn. 8) | $A\mathbb{F}$ (Eqn. 4) | TCL $A\mathbb{A}$ | TCL $A\mathbb{F}$ |
|---|---|---|---|---|---|---|---|---|---|---|---|---|---|
| L2P++ | 79.16 | 0.62 | 5.40 | 14.05 | 62.82 | 4.09 | 2.97 | 2.73 | 2.98 | 19.42 ± 0.09 | 5.49 ± 0.37 | 68.68 ± 0.67 | 10.25 ± 0.77 |
| DualPrompt | 82.92 | 15.27 | 12.19 | 33.71 | 80.12 | 7.65 | 3.17 | 3.47 | 3.23 | 26.86 ± 0.40 | 3.54 ± 0.27 | 76.57 ± 0.23 | 2.57 ± 0.26 |
| CODA-Prompt | 88.06 | 18.09 | 12.16 | 53.95 | 95.65 | 19.42 | 6.83 | 6.34 | 5.62 | 34.01 ± 0.99 | 8.0 ± 0.72 | 75.83 ± 0.60 | 6.25 ± 0.60 |
| S-Prompts | 87.38 | 88.53 | 64.05 | 72.17 | 98.53 | 99.65 | 96.63 | 45.49 | 58.07 | 78.95 ± 0.07 | 0.32 ± 0.32 | 79.71 ± 0.08 | 0.0 ± 0.0 |
| EWC | 79.15 | 85.62 | 47.03 | 64.57 | 90.65 | 54.56 | 55.26 | 34.87 | 54.49 | 62.91 ± 0.81 | 19.82 ± 0.69 | 75.00 ± 0.67 | 6.33 ± 0.30 |
| L2 Regularization | 82.59 | 65.57 | 51.57 | 26.83 | 95.62 | 98.50 | 83.04 | 33.80 | 51.67 | 65.47 ± 0.33 | 7.23 ± 0.51 | 69.63 ± 0.17 | 3.00 ± 0.47 |
| Experience Replay | 56.68 | 6.23 | 33.45 | 74.33 | 7.00 | 0.02 | 64.25 | 27.03 | 37.55 | 34.06 ± 0.69 | 44.89 ± 0.72 | 54.55 ± 2.67 | 22.67 ± 3.01 |
| Our CHEEM | **88.56** | **95.63** | **75.05** | **83.81** | **99.15** | 99.64 | 90.92 | **55.53** | 56.42 | **82.74** ± 0.54 | 0.33 ± 0.00 | **84.65** ± 0.33 | 0.0 ± 0.0 |

## 3 EXPERIMENTS

**Data**: We use the VDD benchmark (Rebuffi et al., 2017a) and the Split ImageNet-R(etention) benchmark introduced in Wang et al. (2022c), derived from (Hendrycks et al., 2021). The VDD benchmark (Fig. 2) consists of 10 tasks with significantly varying distributions of classes per task. The vanilla ImageNet-R dataset consists of 200 classes and 30k images in total. The Split ImageNet-R benchmark is constructed by randomly splitting the ImageNet-R dataset into 10 tasks each of which has 20 classes. To test the effects of imbalance class distributions in streaming tasks, we construct another Split ImageNet-R benchmark consisting of 6 tasks with imbalance number of classes (5, 10, 15, 20, 50, 100) randomly sampled from the 200 total classes without replacement for each task. More details are provided in Appendix. E.

**Baselines of ExfCCL.** We compare with S-Prompts (Wang et al., 2022a), Learning to Prompt (L2P) Wang et al. (2022d), DualPrompt Wang et al. (2022c) and CODA-Prompt Smith et al. (2023a). On the VDD benchmark, we also test two regularization based methods, the L2 Parameter Regularization (Smith et al., 2023b) and the EWC (Kirkpatrick et al., 2017), and a strong baseline of Experience Replay (with iCARL (Rebuffi et al., 2017b) for buffer updates) using the total buffer size 20k and 10 exemplars per class. For EWC, L2 and iCarl, we freeze the backbone model and only keep the output projection layers (where CHEEM is placed) in the MHSA blocks trainable for better understanding the advantages of structurally updating them in our CHEEM. More details are in Appendix F and G.

**Metric**: In addition to average forgetting (Eqn. 4), we compare average accuracy in ExfCCL setting,

$$A\mathbb{A} = \frac{1}{N} \sum_{t=1}^{N} \text{Acc}(T_t; f_{1:N}, C_t). \tag{8}$$

### 3.1 RESULTS ON THE VDD BENCHMARK

Table 2 shows the results. We use the ImageNet-1k (the 1st task in the VDD) trained ViT-B/8 as the base model. For fair comparisons with prompting based baselines, we evaluate average accuracy and forgetting on the remaining 9 tasks in the VDD. **Our proposed CHEEM obtains significantly better performance than all the baselines.** We analyze the results as follows.

**i) The Advantage of Task-Aware Dynamic Backbone Over Frozen Backbone: CHEEM vs S-Prompts.** CHEEM adopts the same task ID inference method (i.e., the external memory) proposed by S-Prompts (Wang et al., 2022a). The improvement by CHEEM with above 3% absolute average accuracy increase clearly shows the advantage of our proposed internal parameter memory for maintaining task-aware dynamic backbones, against the method of prepending retrieved task-specific prompts in S-Prompts. Table 2 also shows a potential for harnessing prompt-based methods in CHEEM. S-Prompts performs better on tasks that are similar to the first ImageNet task and have less training data such as Flwr (918 training images) and DTD (1692 training images).

**ii) Explicit Task ID Inference: With vs Without.** The three baselines, L2P (Wang et al., 2022d), DualPrompt (Wang et al., 2022c) and CODA-Prompt (Smith et al., 2023a), use a shared head classifier in inference, which, as aforementioned in the introduction, suffers from the discrepancy between training and testing. Due to the significant imbalance class distributions in the VDD, their average accuracy undergo catastrophic drops (19.42% by L2P, 26.86% by DualPrompt and 34.01% by CODA-Prompt, vs 78.95% by S-Prompts and 82.74% by our CHEEM).

To understand the effects of head classifiers better, we also compare the performance when task IDs are provided in inference (i.e., Task-incremental CL, TCL), so the three methods can use task-specific

head segments of the shared classifier, and both S-Prompts and our CHEEM do not need the external task-centroid memory. The last two columns of Table 2 show the results. *We have three observations:*

- The performance of L2P, DualPrompt and CODA-Prompt are drastically boosted (e.g., from 34.01% to 75.83% by CODA-Prompt), showing the importance of explicitly inference task IDs for imbalance class distributions of streaming tasks in continual learning. Our CHEEM under CCL is still significantly better than the three methods under TCL (by more than 6%), **which reinforces the importance of the task-aware dynamic backbone learned via our CHEEM**.
- The performance difference between CCL and TCL by our CHEEM is small, around 2%, showing that the task ID inference method (Wang et al., 2022a) is effective in handling the imbalance class distributions in the VDD benchmark.
- In terms of average forgetting under CCL and TCL, we should compare the ratio between average forgetting and average accuracy, rather than the average forgetting rates in isolation, for which the first three methods undergo severer forgetting under CCL as intuitively expected. Our CHEEM does not have forgetting under TCL.

**iii) Stability vs Plasticity of the Backbone.** To further show the advantage of our CHEEM learning to update the feature backbone from the base model, we compare with EWC (Kirkpatrick et al., 2017), L2 Regularization (Smith et al., 2023b) and Experience Replay (Rebuffi et al., 2017b). The three methods regularize the weights, including the shared head classifier, in learning new tasks using different formulations. Comparing with L2P, DualPromt and CODA-Prompt, under CCL, the regularization based methods work better, showing the negative effects of the discrepancy of handling the shared head in training and testing by the three prompting based methods, as well as a potential of integrating them. Comparing with S-Prompts, the plasticity of updating the entire backbone with regularization is less effective than the plasticity introduced by prompts while keeping the backbone frozen in S-Prompts. Comparing with our CHEEM, the three regularization based methods suffer from significant forgetting, highlighting the balanced stability and plasticity via our CHEEM.

Table 3: Results on two constructed Split ImageNet-R benchmarks, averaged across 3 different task orders. The base model is ImageNet-1k trained ViT-B/16 sourced from (Wightman, 2019).

| Method | Imbalanced (6 tasks with (5, 10, 15, 20, 50, 100) classes per task) | | | | Balanced (10 tasks with 20 classes per task) | | | |
|---|---|---|---|---|---|---|---|---|
| | $A\mathbb{A}$ (Eqn. 8) | $A\mathbb{F}$ (Eqn. 4) | TCL $A\mathbb{A}$ | TCL $A\mathbb{F}$ | $A\mathbb{A}$ (Eqn. 8) | $A\mathbb{F}$ (Eqn. 4) | TCL $A\mathbb{A}$ | TCL $A\mathbb{F}$ |
| L2P++ | $64.44 \pm 1.38$ | $8.62 \pm 4.02$ | $88.21 \pm 0.28$ | $0.50 \pm 0.36$ | $73.01 \pm 0.57$ | $5.80 \pm 0.29$ | $89.08 \pm 0.38$ | $0.48 \pm 0.02$ |
| DualPrompt | $67.20 \pm 1.38$ | $6.95 \pm 3.96$ | $89.46 \pm 0.57$ | $0.45 \pm 0.24$ | $73.01 \pm 0.55$ | $3.35 \pm 0.36$ | $90.07 \pm 0.19$ | $0.29 \pm 0.19$ |
| CODAPrompt | $\mathbf{67.66} \pm 3.09$ | $7.06 \pm 2.97$ | $90.69 \pm 0.44$ | $0.46 \pm 0.23$ | $\mathbf{76.48} \pm 0.25$ | $5.04 \pm 0.12$ | $91.86 \pm 0.34$ | $0.45 \pm 0.10$ |
| Our CHEEM | $66.97 \pm 0.43$ | $8.44 \pm 3.55$ | $\mathbf{91.48} \pm 0.63$ | $0.0 \pm 0.0$ | $64.72 \pm 0.28$ | $8.73 \pm 0.41$ | $\mathbf{92.47} \pm 0.64$ | $0.0 \pm 0.0$ |

## 3.2 Results On the Split ImageNet-R Benchmark

The ImageNet-R dataset is created to challenge models trained using ImageNet. So, by design, the external memory of our CHEEM that is based on ImageNet-1k trained base model will not work well. Table 3 shows the results which we analyze as follows:

**i) Randomly Assigned and *Imbalanced* Classes in Streaming Task Challenge All of ExfCCL Methods We Test.** Under CCL, the three prompting based methods suffer from catastrophic drop of performance from balanced to imbalanced settings (e.g., from $76.48 \pm 0.25\%$ to $67.66 \pm 3.09\%$ by CODA-Prompt, see Fig. 5 and analyses in Appendix A ). Our CHEEM is on-par with DualPrompt and CODA-Prompt. Our CHEEM is also stable from balanced to imbalanced scenarios. Under TCL, all methods obtain significant performance boost, and our CHEEM is better than all others. Similarly, we envision that leveraging the dynamically learned backbone by our CHEEM in constructing the external memory will be a promising direction to be studied in future work.

**ii) Randomly Assigned Yet *Balanced* Classes in Streaming Tasks Challenge the Task-Centroid based Task ID Inference.** On the one hand, for streaming tasks with balanced but randomly sampled classes per task (without replacement), our CHEEM under CCL has the worst performance (64.72% vs 76.48% by CODA-Prompt), which was caused by the low average precision of task ID inference (see Fig. 6 and analyses in Appendix B). Our CHEEM under TCL obtains the best performance (92.47%), which shows that the learned task-aware backbone is expressive. Overall, task centroids computed using the CLS tokens in the base model are not able to distinguish tasks from each other with high accuracy. One potential solution is to leverage the dynamically learned backbone by our CHEEM in constructing the external memory, considering its superior performance under TCL, at the expense of more costly task ID inference. On the other hand, *although it is interesting to test continual learning approaches using the Split ImageNet-R benchmark, the random composition of*

*classes in a task is not natural in comparison with scenarios in natural human learning. Unlike the Split ImageNet-R, the VDD benchmark may suit continual learning better from perspective the perspective of real-world scenarios, for which our CHEEM works the best.*

## 4 VISUALIZATION AND ABLATION STUDIES

Due to space limit, we present visualizations of the learned CHEEM (examples like Fig. 2) in Appendix C. We also conduct ablation studies including: **(i)** The proposed HEE sampling significantly outperforms pure exploration (PE) sampling in Appendix D.1. **(ii)** The proposed HEE-empowered SPOS NAS significantly outperforms the DARTS (Liu et al., 2019a) based NAS used in the learn-to-grow method (Li et al., 2019) in Appendix D.2. **(iii)** The proposed internal memory learning method outperforms three state-of-the-art methods: SupSup (Wortsman et al., 2020), EFT (Verma et al., 2021) and LL (Ge et al., 2023) under the task-to-task transfer learning setting in Appendix D.3. And, more are in Appendix D.3, D.4 and D.6.

## 5 RELATED WORK

*Experience Replay Based approaches* aim to retain some exemplars, in the form of either raw data or latent features, from the previous tasks and replay them to the model along with the data from a new task (Aljundi et al., 2019b; Rebuffi et al., 2017b; Aljundi et al., 2019a; Balaji et al., 2020; Bang et al., 2021; Chaudhry et al., 2021; Pham et al., 2021), or *generative replay methods* (Shin et al., 2017; Cong et al., 2020) which replay exemplars sampled from the learned generators along with the data from the current task. For exemplar-free continual learning, *Regularization-based Approaches* explicitly prevent model parameters from deviating too far from their stable values learned on previous tasks when learning a new task (Aljundi et al., 2018; 2019c; Douillard et al., 2020; Nguyen et al., 2018; Kirkpatrick et al., 2017; Li & Hoiem, 2018; Zenke et al., 2017; Schwarz et al., 2018).

*Dynamic Models* aim to use different parameters for each task to eliminate the use of stored exemplars. Dynamically Expandable Network (Yoon et al., 2018) adds neurons to a network based on learned sparsity constraints and heuristic loss thresholds. PathNet (Fernando et al., 2017) and RPSNet (Rajasegaran et al., 2019) learn task-specific submodules or paths. Progressive Neural Networks (Rusu et al., 2016) learn a new network per task and adds lateral connections to the previous tasks' networks. The L2G (Li et al., 2019) uses Differentiable Architecture Search (DARTS) (Liu et al., 2019a) to determine if a layer can be reused, adapted, or renewed for a task, which is tested for ConvNets and the learning-to-grow operations are applied uniformly at each layer in a ConvNet. Our method is motivated by the L2G method, but with significant differences. NAS has also been explored by (Gao et al., 2023a) for continual learning on small scale tasks. Approaches that learn task-specific components on top of a backbone include (Ge et al., 2023; Verma et al., 2021; Wortsman et al., 2020; Abati et al., 2020).

Recently, there has been increasing interest in continual learning using Vision Transformers (Wang et al., 2022d;c; Xue et al., 2022; Ermis et al., 2022; Douillard et al., 2022; Pelosin et al., 2022; Yu et al., 2021; Li et al., 2022; Iscen et al., 2022; Wang et al., 2022a;b; Mohamed et al., 2023; Gao et al., 2023b; Zhou et al., 2023). *Prompt Based approaches* learn external parameters appended to the data tokens that encode task-specific information useful for classification (Wang et al., 2022d;a; Douillard et al., 2022; Smith et al., 2023a; Jung et al., 2023; Tang et al., 2023; Zhou et al., 2024). Our proposed method is complementary to prompting-based based approaches.

## 6 CONCLUSION

This paper presents a method of transforming Vision Transformers (ViTs) for exemplar-free class-incremental continual learning (ExfCCL), dubbed **CHEEM** (Continual Hierarchical-Exploration-Exploitation Memory). Our CHEEM consists of the external (task-centroid) memory and the internal (parameter) memory. The former is for task ID inference for test data based on clustered task centroids in training. The latter is realized by a proposed Hierarchical-Exploration-Exploitation (HEE) sampling based neural architecture search algorithm. The external and internal memory are maintained in a decoupled way. Our CHEEM is tested on two challenging benchmarks, the VDD benchmark and the continual ImageNet-R benchmark. It obtains state-of-the-art performance on the VDD outperforming the prior art by a large margin, with sensible CHEEM strutures continually learned. It obtains on-par performance on the imbalanced ImageNet-R benchmark, while performs worse on the balanced one, for which we analyze the reason thoroughly due to the external memory.

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

APPENDIX

OUTLINE

In this Appendix, we elaborate on the following aspects that are not presented in the submission due to the space limit:

- **Section A – Analyses of head classifier design choices** in continual learning.
- **Section B – Analysis of the discriminative capacity of centroids for task ID inference**
- **Section C – More examples of learned CHEEM** by our proposed hierarchical exploration-exploitation (HEE) sampling scheme and by the vanilla pure exploration (PE) sampling scheme.
- **Section D - Ablation Studies**:
    - **Section D.1 – Comparing Pure Exploration vs. our propoed Hierarchical Exploration-Exploitaiton sampling scheme for SPOS NAS**
    - **Section D.2 – Comparing proposed HEE empowered SPOS NAS with DARTS**: We compare with DARTS, which has been used by Learn-to-Grow and show that our proposed method outperforms L2G with DARTS.
    - **Section D.3 – Can CHEEM be used for task-to-task transfer?**
    - **Section D.4 – Can we use other components in ViTs as CHEEM?**: We compare with the Query/Key/Value layer and the FFN layer, and verify the effectiveness of the proposed identification in the main paper.
    - **Section D.5 – How is the** `Adapt` **operation implemented?**: We elaborate two implementation methods of the `Adapt` operation and compare their performance.
    - **Section D.6 - Effects of task orders.**
- **Section E – Details of the two benchmarks** tested in the experiments: the Visual Domain Decathlon (VDD) (Rebuffi et al., 2017a) benchmark and the ImageNet-R (Hendrycks et al., 2021) benchmark.
- **Section F – The base model and its training details**: the Vision Transformer (ViT) model specification (ViT-B/8) and how it is initially training on the ImageNet in our experiments.
- **Section G – Experimental settings and training hyperparameters in our implementation** on the VDD benchmark and the 5-dataset benchmark.
- **Section H** - Details of Modifying SupSup, EFT and LL on the VDD Benchmark to work with Vision Transformers:

# A TASK-SPECIFIC HEAD SELECTION VIA TASK ID INFERENCE VS SHARING A GROWING, COMMON TASK HEAD

As aforementioned in the introduction of main text, explicitly inferring task IDs from the external memory for test data is to select the task head classifiers as done in (Wang et al., 2022a), while also enabling selecting task-specific feature backbones (exploited in our CHEEM). An alternative approach is to share a growing, common task head as done in (Wang et al., 2022d;c; Smith et al., 2023a). The former can suffer from low average precision, while for the latter we observe serious instability caused by imbalanced classes in stream tasks due to the discrepancy in training and testing (see Table 2 and Table 3).

To show the instability, let $\mathcal{C} = (z_1, z_2, \cdots, z_K)$ be the logits output of the common head. During training, the task ID for each $z_k \in \mathcal{C}$ is known, and the logits of previous tasks are masked out (using $-\infty$) before computing the softmax (i.e., task-specific heads are used in training). During testing, the label for a testing sample is predicated by $\arg\max_{k \in [1,K]} z_k$, across the entire head. This discrepancy between training and testing imposes a very strong assumption of the stability of logits in training and testing, which entails that the task-specific head segment for a testing example will underlyingly resemble the 1-hot vector, such that it can hold across the entire head. Although (Wang et al., 2022d;c; Smith et al., 2023a) show strong performance using the same number of classes per task (20) in ImageNet-R (Hendrycks et al., 2021), we observe that slightly varying the number of classes per task, while keeping everything else the same, can lead to around 10% drop of performance in ImageNet-R. For more challenging task-class distributions as in the VDD (Fig. 2), (Wang et al., 2022d;c; Smith et al., 2023a) have a catastrophic drop of performance.

To analyze this further, we probe the model after the training on the final task is complete. For each task, we first make predictions using the task specific part of the head (i.e., the task-incremental setting) and calculate the entropy of predicted probabilities of all the samples with the task-specific head. Then, we make predictions with the full head (i.e., the class-incremental setting), including all the classes in the previous tasks, and track samples for which the logit with the maximum values changes to one of the classes from the previous tasks, which we refer to as **switched samples**.

As shown in Fig. 5, we observe that the entropy for the switched samples is consistently higher than those of the un-switched samples. Furthermore, the variance of the the entropy consistently goes down as the tasks progress, which indicates that for later tasks, only the predictions that are highly confident remain unswitched. However, the entropy for the switched samples shows a large variance. This indicates that sharp predictions are not a sufficient condition to avoid switching.

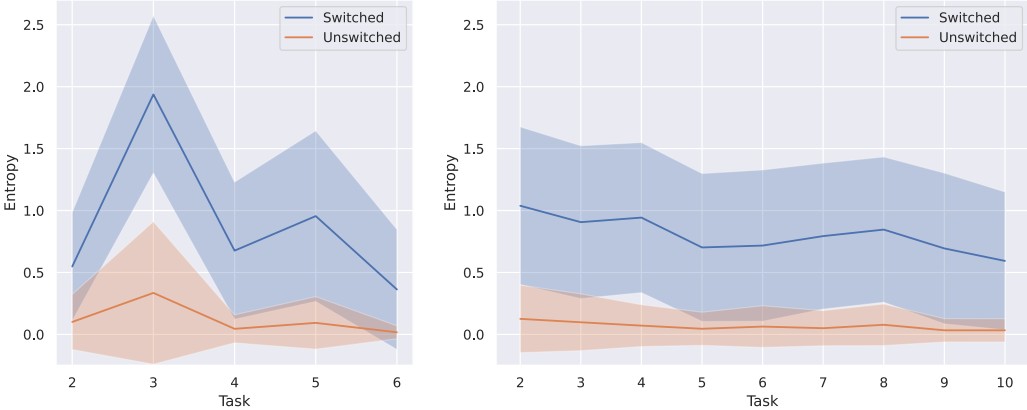

Figure 5: Mean entropy for the switched and unswitched samples from each task on the Split ImageNet-R benchmarks using CODA-Prompt. *Left:* Imbalanced ImageNet-R with varying number of classes (5,10,15,20, 50, 100) per task and 6 tasks in total. *Right:* Balanced ImageNet-R with same number (20) of classes per task and 10 tasks in total.

## B    THE PROS AND CONS OF BASE MODEL COMPUTED TASK CENTROIDS AS THE EXTERNAL MEMORY FOR TASK ID INFERENCE

To understand effects of the external task-centroid memory we adopted from S-Prompts (Wang et al., 2022a), we analyze the separability of the K-Mean clusters of different tasks on the VDD (Rebuffi et al., 2017a) (see results in Table 2) and Balanced ImageNet-R benchmarks (see results in Table 3) by visualizing them using t-SNE (van der Maaten & Hinton, 2008) in Fig. 6.

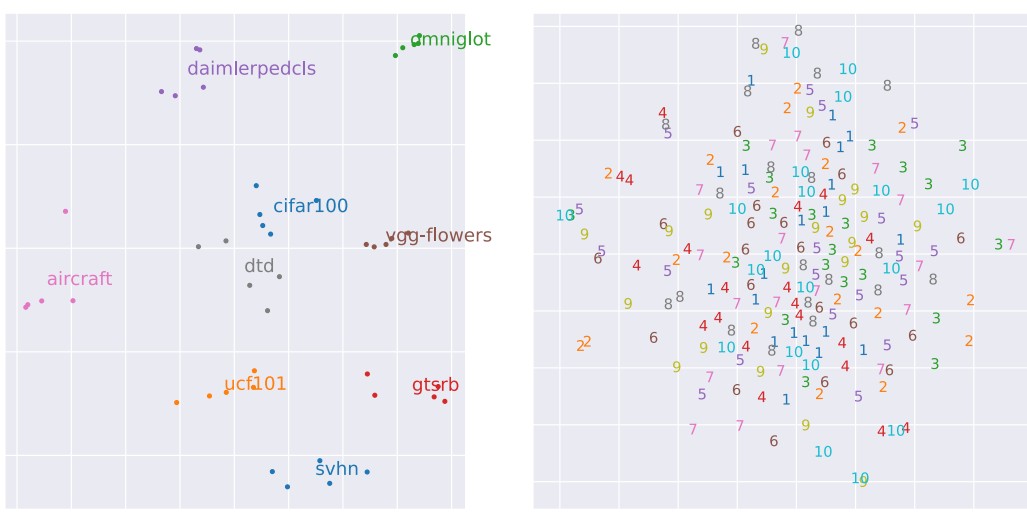

**VDD**                    **Balanced ImageNet-R**

Figure 6: The t-SNE visualization of task centroids using the frozen base model `CLS` tokens of training examples of each task. We run $K$-Mean on the VDD benchmark with $K = 5$ per task. We use the mean `CLS` token per class (20 centroids per task) on the balanced Imagenet-R benchmark, which we found works better.

For our proposed CHEEM, we can observe that the external task-centroid memory is effective on the VDD, providing sufficiently high recall rates for the internal memory of CHEEM and resulting in overall high average accuracy and negligible average forgetting. On the contrary, the external task-centroid memory of the balanced Imagenet-R benchmark explains why our CHEEM performs much worse under CCL. We note that the external memory is constructed based on the frozen base model. As discussed in the main text, a potential improvement is to leverage dynamically updated backbones via the internal memory of our CHEEM in computing the task centroids at the expense of increased task ID inference cost. That is also to shift from our current decoupled external and internal memory design to integrative / coupled external and internal memory.

Together with the experimental results that the prompting based methods (L2P, DualPrompt and CODA-Prompt) can handle balanced ImageNet-R better in terms of preventing switched samples as analyzed in Appendix A, and that regularized based method (EWC and L2 Regularization in Table 2) can handle VDD better than prompting based methods, we envision there are opportunities of exploring the integration between them and the internal memory of our CHEEM for more robust ExfCCL across scenarios including both VDD and balanced/imbalanced ImageNet-R.

## C    EXAMPLES OF CHEEM LEARNED SEQUENTIALLY AND CONTINUALLY

Fig. 7 shows the CHEEM learned sequentially and continually via the proposed HEE-based NAS on the VDD benchmark (Rebuffi et al., 2017a) with three different random seeds.

As comparisons, Fig. 8 shows the structure updates learned using the vanilla PE-based NAS. It can be seen that pure exploration does not reuse components from similar tasks. The pure exploration based method adds many unnecessary `Adapt` and `New` operations even though the tasks are similar (e.g., ImNet → C100), verifying the effectiveness of the proposed sampling method. While the pure

exploration scheme adds many `Skip` operations, thereby reducing the overall FLOPs, the average accuracy is low by a large margin, about 6%. This shows that the pure exploration scheme cannot learn to choose operations in a task synergy aware way.

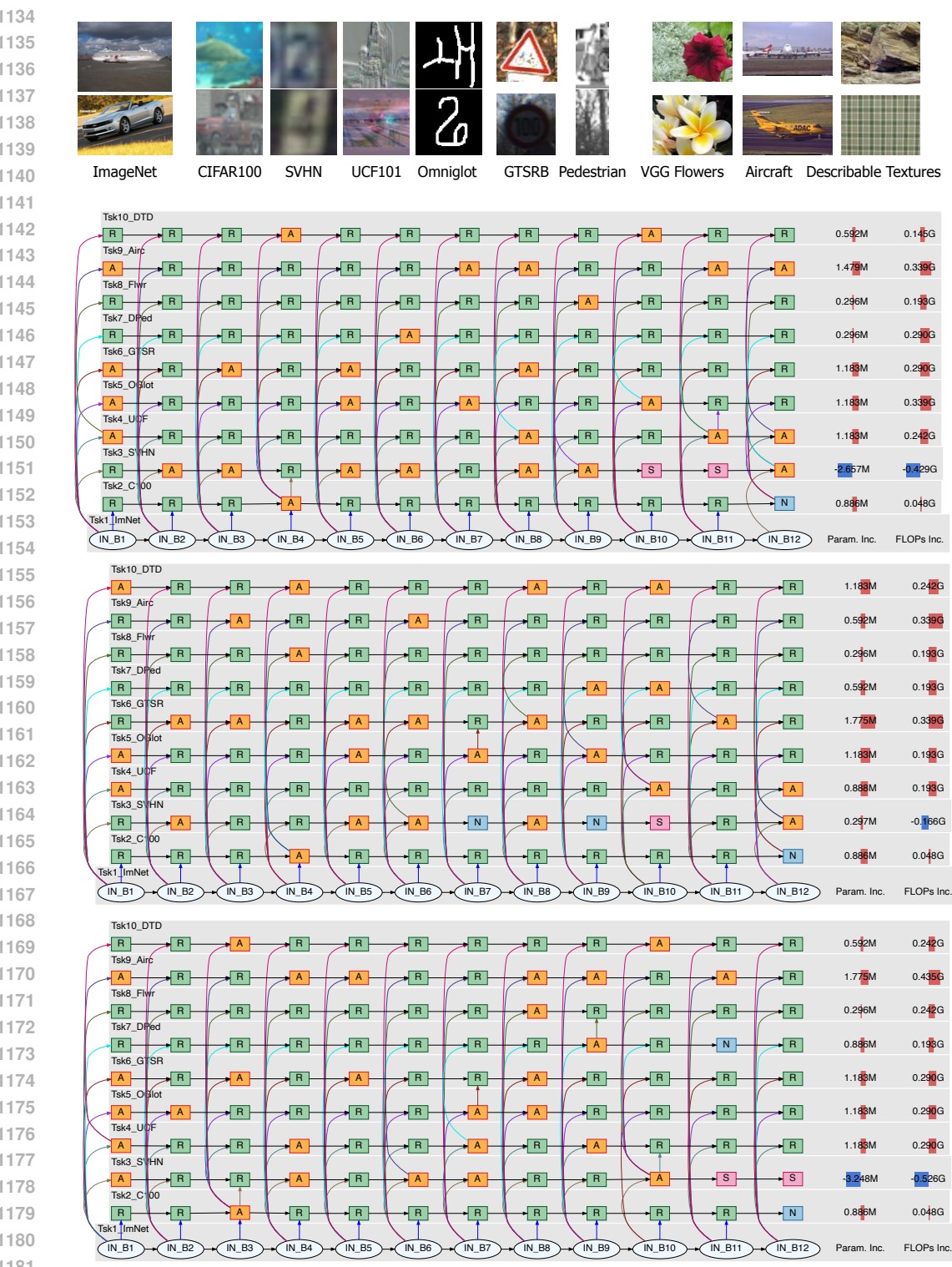

Figure 7: Examples of the task-synergy memory (CHEEM) learned on the VDD benchmark (Rebuffi et al., 2017a) with the task sequence shown in the top **using our proposed HEE-based NAS** and three different random seeds. The overall performance is reported in Table 2 in the main paper. S, R, A and N represent Skip, Reuse, Adapt and New respectively. The last two columns show the number of new task-specific parameters and added FLOPs respectively, in comparison with the first task, ImNet model. Overall, the learned task synergies make intuitive sense and remain relatively stable across different random seeds.

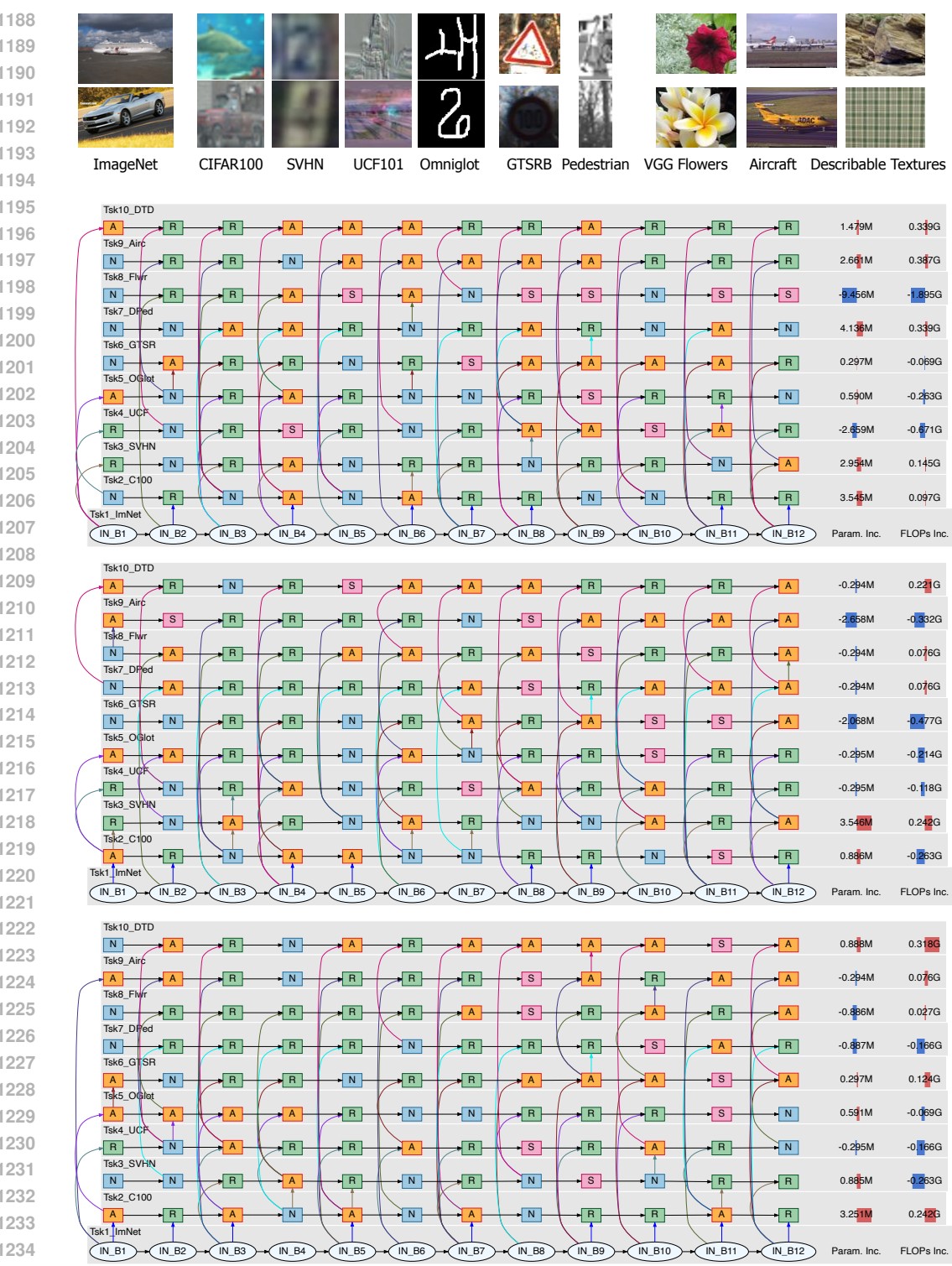

Figure 8: Examples of the task-synergy memory (CHEEM) learned on the VDD benchmark (Rebuffi et al., 2017a) with the task sequence shown in the top **using the vanilla PE-based NAS** and three different random seeds. S , R , A and N represent Skip, Reuse, Adapt and New respectively. The overall performance is reported in Table 4 in this appendix and our proposed HEE-based NAS significantly improves the performance by an absolute 6% average accuracy. In terms of the learned CHEEM, the PE-based NAS leads to much more New operations, which shows it is less effective in terms of leveraging task synergies.

# D  ABLATION STUDIES

## D.1  HIERARCHICAL EXPLORATION AND EXPLOITATION (HEE) VS. PURE EXPLORATION (PE)

We verify the effectiveness of our HEE sampling (Table 4), which is significantly better by a large margin, 6% absolute average accuracy increase.

Table 4: The effectiveness of our proposed hierarchical exploration and exploitation sampling empowered SPOS NAS (Fig. 4). The structure updates learned by the PE strategy are visualized in Fig. 8 in the Appendix.

| Method | C100 | SVHN | UCF | OGlt | GTSR | DPed | Flwr | Airc. | DTD | Avg. Accuracy | Avg. Forgetting |
|---|---|---|---|---|---|---|---|---|---|---|---|
| PE | 80.34 | 94.75 | 72.63 | 81.43 | 99.13 | 99.59 | 72.45 | 38.27 | 37.22 | 75.09 ± 0.81 | 0.32 ± 0.00 |
| HEE | 88.56 | 95.63 | 75.05 | 83.81 | 99.15 | 99.64 | 90.92 | 55.53 | 56.42 | 82.74 ± 0.54 | 0.33 ± 0.0 |

## D.2  COMPARISON WITH DARTS

Table 5 shows the effectiveness of the proposed HEE empowered SPOS NAS against DARTS, as used in the original learn-to-grow method. Our proposed HEE empowered SPOS NAS outperforms DARTS, and the more advanced $\beta$-DARTS (Ye et al., 2022) by a large margin. We compare with Learn-to-Grow under a task-incremental continual learning (TCL) setting as used in the original formulation of L2G by (Li et al., 2019).

Table 5: The effectiveness of our proposed HEE sampling empowered SPOS NAS (Fig. 4). The two L2G variants use the same ViT-B base model as our HEE.

| Method | C100 | SVHN | UCF | OGlt | GTSR | DPed | Flwr | Airc. | DTD | Avg. Accuracy |
|---|---|---|---|---|---|---|---|---|---|---|
| L2G (DARTS) | 88.47 | 85.20 | 79.22 | 80.19 | 99.28 | 98.06 | 76.14 | 39.29 | 46.01 | 77.45 ± 2.41 |
| L2G ($\beta$-DARTS) | 88.95 | 94.73 | 75.31 | 79.76 | 99.84 | 99.76 | 78.86 | 34.50 | 47.09 | 78.14 ± 0.54 |
| HEE | 90.93 | 95.96 | 80.74 | 83.25 | 99.94 | 99.96 | 94.12 | 58.90 | 60.05 | 84.65 ± 0.33 |

## D.3  RESULTS BY TASK-TO-TASK TRANSFER BASED CONTINUAL LEARNING

To evaluate the transfer learning capabilities of CHEEM, we compare with three state-of-the-art methods: Supermasks in Superposition (SupSup) (Wortsman et al., 2020), Efficient Feature Transformation (EFT) (Verma et al., 2021) and Lightweight Learner (LL) (Ge et al., 2023) that modify the backbone model separately for each taskm, which we refer to as a task-to-task transfer setting. We modify all the methods to work with ViTs. We provide the details of the modifications in Section H. All three methods are fine-tuned on a new task for 50 epochs. For CHEEM, we use 50 epochs for the supernet training and 30 epochs for fine-tuning.

Table 6 demonstrates that CHEEM achieves the highest average accuracy on the VDD benchmark. Although its performance on individual tasks may be lower, CHEEM exhibits robustness to domain variations, as reflected in its overall average accuracy. In contrast, other methods show varying performance depending on the domain. SupSup performs well on tasks that are significantly different from the base ImageNet task, such as UCF 101, Omniglot, and SVHN, but performs poorly on tasks closely related to ImageNet. Conversely, EFT and LL excel on tasks similar to ImageNet but struggle with out-of-distribution tasks. Despite its lower individual task performance, CHEEM's consistent performance across diverse downstream tasks highlights its adaptability and makes it more general.

Table 6: Results on the VDD benchmark (Rebuffi et al., 2017a) using ViT-B/8 (Dosovitskiy et al., 2021) under the task-to-task transfer based continual learning protocol. The learned CHEEM are visualized in Fig. 10 in the Appendix.

| Method | ImNet | C100 | SVHN | UCF | OGlt | GTSR | DPed | Flwr | Airc. | DTD | Avg. Accuracy |
|---|---|---|---|---|---|---|---|---|---|---|---|
| SupSup | 82.65 | 89.96 | **96.05** | 81.68 | **84.60** | **99.97** | 99.97 | 78.76 | 44.18 | 51.60 | 81.14 ± 0.04 |
| EFT | 82.65 | 91.86 | 93.51 | 73.89 | 75.62 | 99.58 | **99.98** | 96.34 | 48.17 | **64.40** | 82.60 ± 0.07 |
| LL | 82.65 | **91.92** | 93.90 | 75.63 | 77.07 | 99.71 | 99.96 | **96.47** | 49.33 | 64.34 | 83.10 ± 0.02 |
| Our CHEEM | 82.65 | 90.93 | 95.96 | 80.74 | 83.25 | 99.94 | 99.96 | 94.12 | **58.90** | 60.05 | **84.65** ± 0.33 |

## D.4 CHEEM Placed at Other ViT Components

Table 7 shows the performance comparisons with other four different components in the ViT (the Query/Key/Value linear projection layer and the FFN block) used in realizing the proposed CHEEM. The Query/Key/Value component as the CHEEM does not perform as well as the Projection component. The FFN block as the CHEEM performs only slightly better than the Projection layer, but at the expense of a much larger parameter cost. This reinforces our identification above.

Table 7: Results of ablation study on other components of the ViT used for realizing the CHEEM. The results have been averaged over 3 different seeds.

| Component | ImNet | C100 | SVHN | UCF | OGlt | GTSR | DPed | Flwr | Airc. | DTD | Avg. Accuracy | Avg. Param. Inc./task (M) |
|---|---|---|---|---|---|---|---|---|---|---|---|---|
| Projection | 82.65 | 90.54 | 96.12 | 75.53 | 83.81 | 99.93 | 99.88 | 91.21 | 55.59 | 59.18 | $83.44 \pm 0.50$ | $1.06 \pm 0.04$ |
| Query | 82.65 | 89.66 | 93.74 | 71.53 | 82.02 | 99.87 | 99.89 | 90.03 | 49.57 | 59.40 | $81.84 \pm 0.32$ | $2.38 \pm 0.12$ |
| Key | 82.65 | 89.29 | 94.77 | 72.25 | 81.86 | 99.86 | 99.90 | 88.86 | 51.72 | 60.46 | $82.16 \pm 0.17$ | $2.41 \pm 0.03$ |
| Value | 82.65 | 84.94 | 95.90 | 75.85 | 84.68 | 99.89 | 99.89 | 86.54 | 48.83 | 55.37 | $81.46 \pm 0.25$ | $1.70 \pm 0.11$ |
| FFN | 82.65 | 91.05 | 96.08 | 76.96 | 85.22 | 99.94 | 99.94 | 93.79 | 56.74 | 59.61 | $84.20 \pm 0.28$ | $2.31 \pm 0.28$ |

## D.5 Implementation Details of the `Adapt` Operation

**How to `Adapt` in a sustainable way?** The proposed `Adapt` operation will effectively increase the depth of the network in a plain way. In the worst case, if too many tasks use `Adapt` on top of each other, we will end up stacking too many MLP layers together. This may lead to unstable training due to gradient vanishing. Shortcut connections (He et al., 2016) have been shown to alleviate the gradient vanishing and exploding problems, making it possible to train deeper networks. We introduce the shortcut connection in adding a MLP `Adapt` operation. We test two different implementations: with shortchut in all the three components (supernet training, target network selection and target network finetuing) versus with shortcut only in target network finetuning (i.e., without shortcut in the NAS including both supernet training and target network selection).

Table 8: Results of the ablation study on the implementation of the `Adapt` operation: with (w/) vs without (w/o) shortcut connection for the MLP `Adapt` layer in NAS. The first two rows are for the sequential and continual paradigm and the last two rows for the task-to-task (T2T) transfer based paradigm.

| Shortcut in NAS | ImNet | C100 | SVHN | UCF | OGlt | GTSR | DPed | Flwr | Airc. | DTD | Avg. Accuracy | Avg. Param. ↑/task (M) | Avg. FLOPs ↑/task (G) |
|---|---|---|---|---|---|---|---|---|---|---|---|---|---|
| w/o | 82.65 | 90.54 | 96.12 | 75.53 | 83.81 | 99.93 | 99.88 | 91.21 | 55.59 | 59.18 | $83.44 \pm 0.50$ | $1.06 \pm 0.04$ | $0.17 \pm 0.01$ |
| w/ | 82.65 | 91.18 | 96.18 | 82.34 | 86.03 | 99.91 | 99.95 | 91.60 | 58.90 | 58.56 | $84.73 \pm 0.19$ | $2.01 \pm 0.18$ | $0.49 \pm 0.13$ |
| w/o (T2T) | 82.65 | 90.93 | 95.96 | 80.74 | 83.25 | 99.94 | 99.96 | 94.12 | 58.90 | 60.05 | $84.65 \pm 0.33$ | $2.61 \pm 0.15$ | $-0.19 \pm 0.09$ |
| w/ (T2T) | 82.65 | 91.24 | 99.25 | 84.14 | 85.99 | 99.97 | 99.95 | 94.64 | 60.34 | 61.63 | $85.68 \pm 0.16$ | $3.23 \pm 0.12$ | $0.01 \pm 0.02$ |

Table 8 shows the performance comparisons on the VDD Benchmark under the continual learning paradigm. In terms of sequentially introduced complexities, a more compact model is learned without the shortcut in the `Adapt` during NAS (supernet training and target network selection) as evidenced by the number of additional parameters and the increase in FLOPs. Using the shortcut in both supernet training and target network selection results in twice the parameter increase, and almost $4\times$ increase in FLOPs. Fig. 9 and Fig. 10 show comparisons of the learned CHEEM by the two implementation methods under the two paradigms respectively.

**Remarks.** We have two remarks as follows.

- We use the more parameter-efficient implementation (i.e., w/o shortcut for the `Adapt` in NAS) in the main paper for both the continual learning and the task-to-task transfer learning paradigms, even though the counterparts have better performance.

- We note that although the T2T paradigm results in larger parameter increase per task, its computational costs are relatively lower due to either more `Skip` operations learned and/or the fact that there is no `Adapt`-on-`Adapt` operations since it is task-to-task transfer based learning.

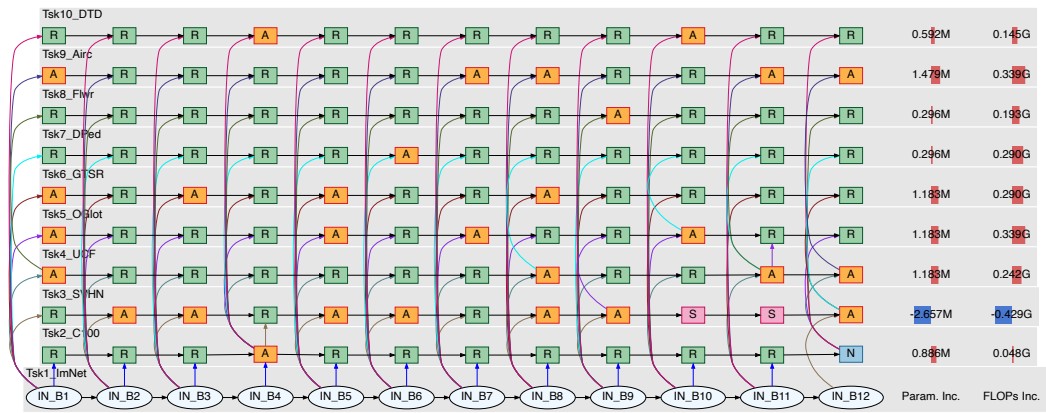

(a) An example of CHEEM learned **without** the shortcut for the MLP `Adapt` layer in NAS (the same one as the 2nd row in Fig. 7).

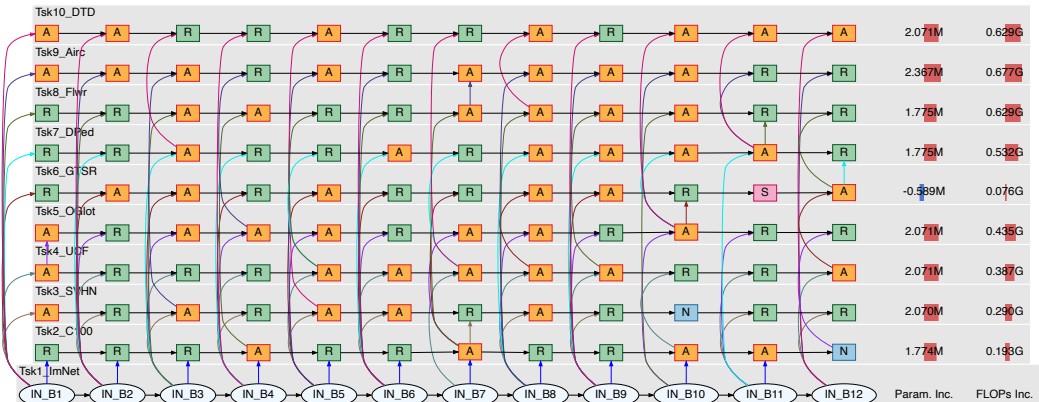

(b) An example of CHEEM learned **with** the shortcut for the MLP `Adapt` layer in NAS. More `Adapt` on top of `Adapt` operations are learned.

Figure 9: Comparisons between CHEEM learned by two different implementations of the MLP `Adapt` operation under the sequential and continual learning paradigm. S , R , A and N represent `Skip`, `Reuse`, `Adapt` and `New` respectively.

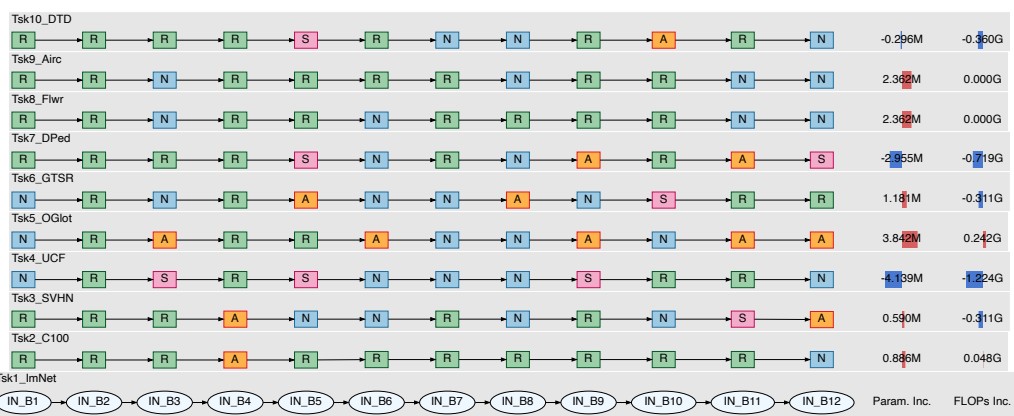

(a) An example of CHEEM learned **without** the shortcut for the MLP `Adapt` layer in NAS.

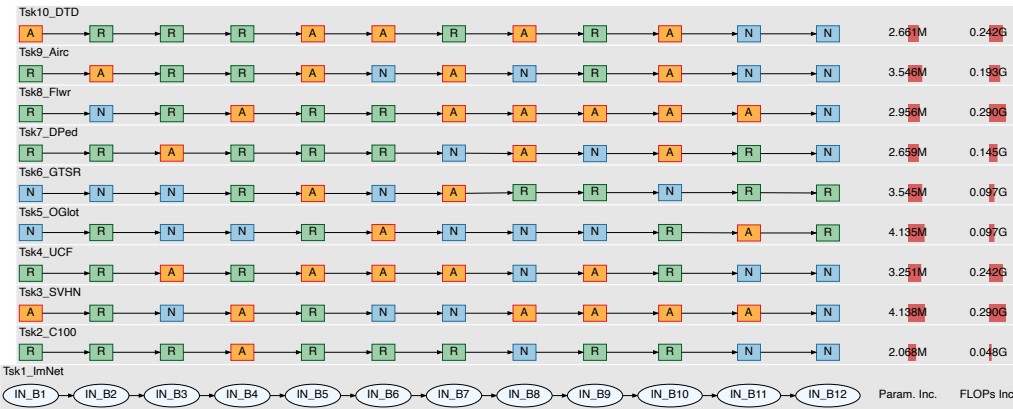

(b) An example of CHEEM learned **with** the shortcut for the MLP `Adapt` layer in NAS. More `Adapt` operations are learned.

Figure 10: Comparisons between CHEEM learned by two different implementations of the `Adapt` operation under the task-to-task (T2T) transfer based lifelong learning setting. Here all the 9 tasks are transferred from the base Tsk1_ImNet model, so we omit the arrows linking the blocks for clarity. S , R , A and N represent `Skip`, `Reuse`, `Adapt` and `New` respectively.

## D.6 EFFECTS OF TASK ORDERS

We investigate the effects of task orders of the 9 tasks in the VDD benchmark. We test four more task sequences in addition to the one presented in the main paper. Overall, The CHEEM learned by our proposed HEE-based NAS achieve similar performance across different task orders, and consistently significantly outperform those learned by the vanilla PE-based NAS. We note that since the task ID inference is based on a frozen backbone, it is independent of the task order. Hence, for simplicity, we evaluate the effect of task order in a task-incremental setting.

Table 9 reports the performance. Fig. 11, Fig. 12, Fig. 13 and Fig. 14 show the learned CHEEM using our proposed HEE-based NAS.

Table 9: Results of ablation study on CHEEM learning with four different task orders using both our proposed HEE-based NAS and the vanilla PE-based NAS. The results have been averaged over 3 different seeds.

| NAS | ImNet | OGlt | UCF | Airc. | Flwr | SVHN | DTD | GTSR | DPed | C100 | Avg. Accuracy | Avg. Param. Inc./task (M) |
|---|---|---|---|---|---|---|---|---|---|---|---|---|
| HEE | 82.65 | 84.32 | 75.27 | 54.32 | 90.29 | 95.83 | 57.89 | 99.92 | 99.72 | 89.96 | $83.02 \pm 0.31$ | $1.25 \pm 0.15$ |
| PE | 82.65 | 77.41 | 70.12 | 39.40 | 64.35 | 94.12 | 37.02 | 99.83 | 99.41 | 70.78 | $73.51 \pm 0.80$ | $2.86 \pm 0.14$ |

| NAS | ImNet | DPed | SVHN | DTD | Airc. | OGlt | C100 | GTSR | Flwr | UCF | Avg. Accuracy | Avg. Param. Inc./task (M) |
|---|---|---|---|---|---|---|---|---|---|---|---|---|
| HEE | 82.66 | 99.94 | 95.83 | 58.56 | 42.43 | 83.55 | 89.98 | 99.95 | 91.99 | 75.67 | $82.06 \pm 1.28$ | $1.42 \pm 0.05$ |
| PE | 82.66 | 99.65 | 95.04 | 45.66 | 35.87 | 77.62 | 71.51 | 99.85 | 66.11 | 63.99 | $73.79 \pm 0.50$ | $2.85 \pm 0.12$ |

| NAS | ImNet | UCF | C100 | OGlt | GTSR | DTD | Flwr | SVHN | DPed | Airc. | Avg. Accuracy | Avg. Param. Inc./task (M) |
|---|---|---|---|---|---|---|---|---|---|---|---|---|
| HEE | 82.66 | 79.73 | 90.75 | 84.93 | 99.90 | 58.14 | 91.27 | 96.05 | 99.89 | 54.06 | $83.74 \pm 0.51$ | $1.37 \pm 0.05$ |
| PE | 82.66 | 74.49 | 74.17 | 78.76 | 99.91 | 41.01 | 70.49 | 94.15 | 99.25 | 37.77 | $75.27 \pm 2.41$ | $2.75 \pm 0.16$ |

| NAS | ImNet | Flwr | UCF | OGlt | GTSR | DPed | C100 | Airc. | DTD | SVHN | Avg. Accuracy | Avg. Param. Inc./task (M) |
|---|---|---|---|---|---|---|---|---|---|---|---|---|
| HEE | 82.66 | 87.52 | 77.17 | 84.20 | 99.92 | 99.80 | 90.30 | 54.49 | 56.83 | 96.03 | $82.89 \pm 0.58$ | $1.41 \pm 0.09$ |
| PE | 82.66 | 72.75 | 76.31 | 78.47 | 99.89 | 99.42 | 70.09 | 34.95 | 39.89 | 93.72 | $74.81 \pm 1.40$ | $2.70 \pm 0.11$ |

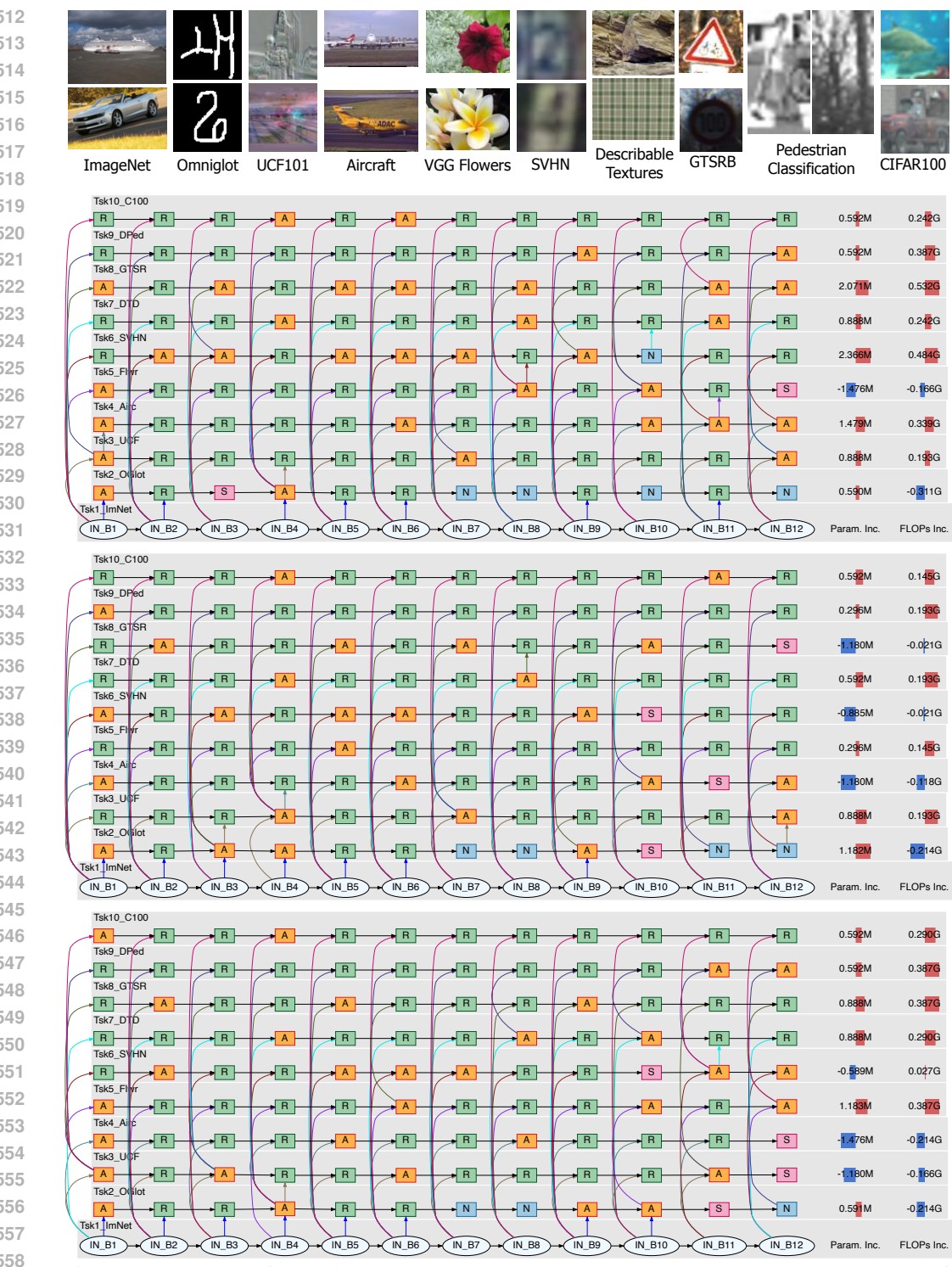

Figure 11: Examples of the task-synergy memory (CHEEM) learned on the VDD benchmark (Rebuffi et al., 2017a) with the task sequence shown in the top **using our proposed HEE-based NAS** and three different random seeds. The overall performance is reported in Table 9. S, R, A and N represent Skip, Reuse, Adapt and New respectively. The last two columns show the number of new task-specific parameters and added FLOPs respectively, in comparison with the first task, ImNet model. Overall, the learned task synergies make intuitive sense and remain relatively stable across different random seeds.

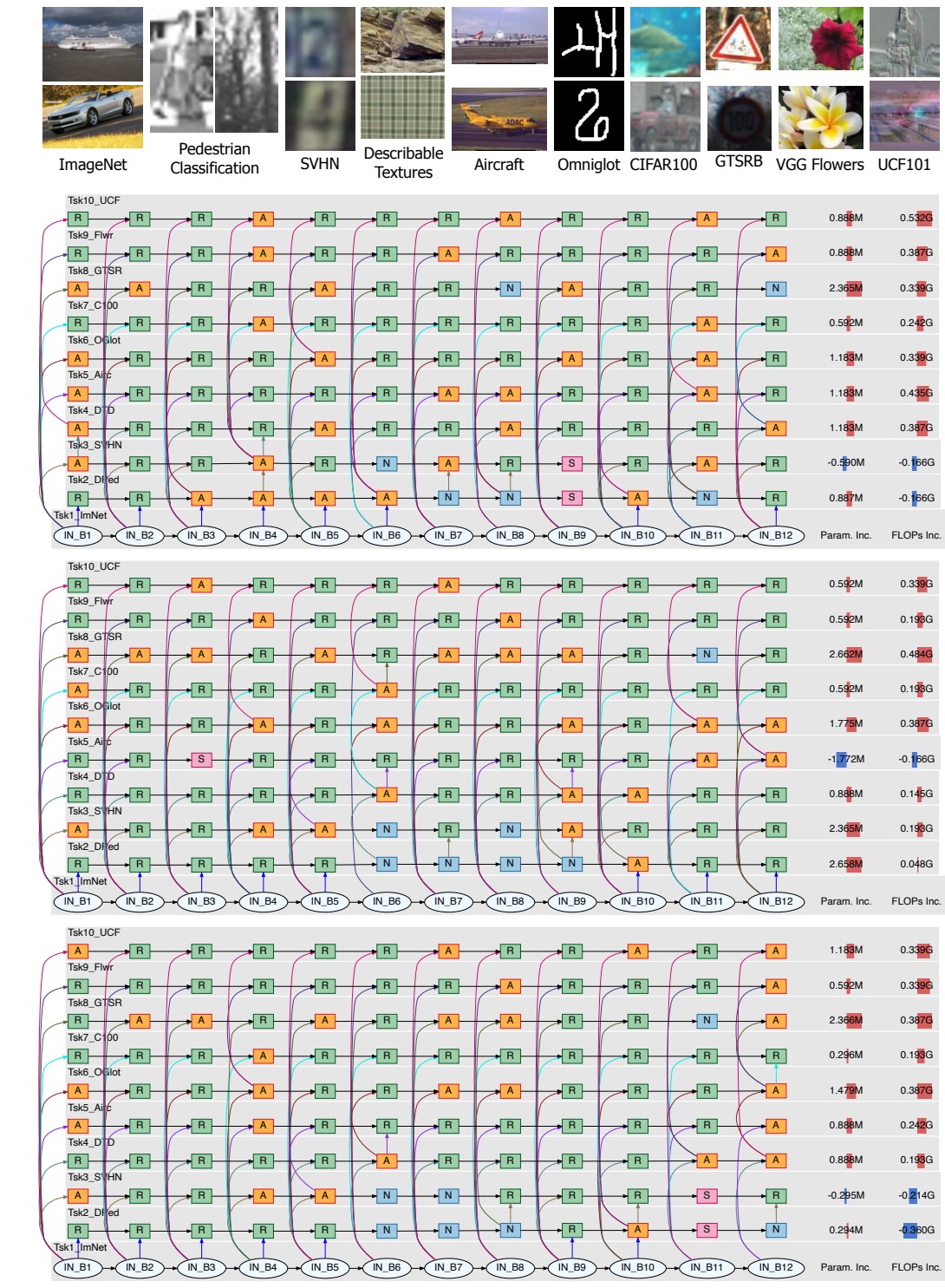

Figure 12: Examples of the task-synergy memory (CHEEM) learned on the VDD benchmark (Rebuffi et al., 2017a) with the task sequence shown in the top **using our proposed HEE-based NAS** and three different random seeds. The overall performance is reported in Table 9. S , R , A and N represent Skip, Reuse, Adapt and New respectively. The last two columns show the number of new task-specific parameters and added FLOPs respectively, in comparison with the first task, ImNet model. Overall, the learned task synergies make intuitive sense and remain relatively stable across different random seeds.

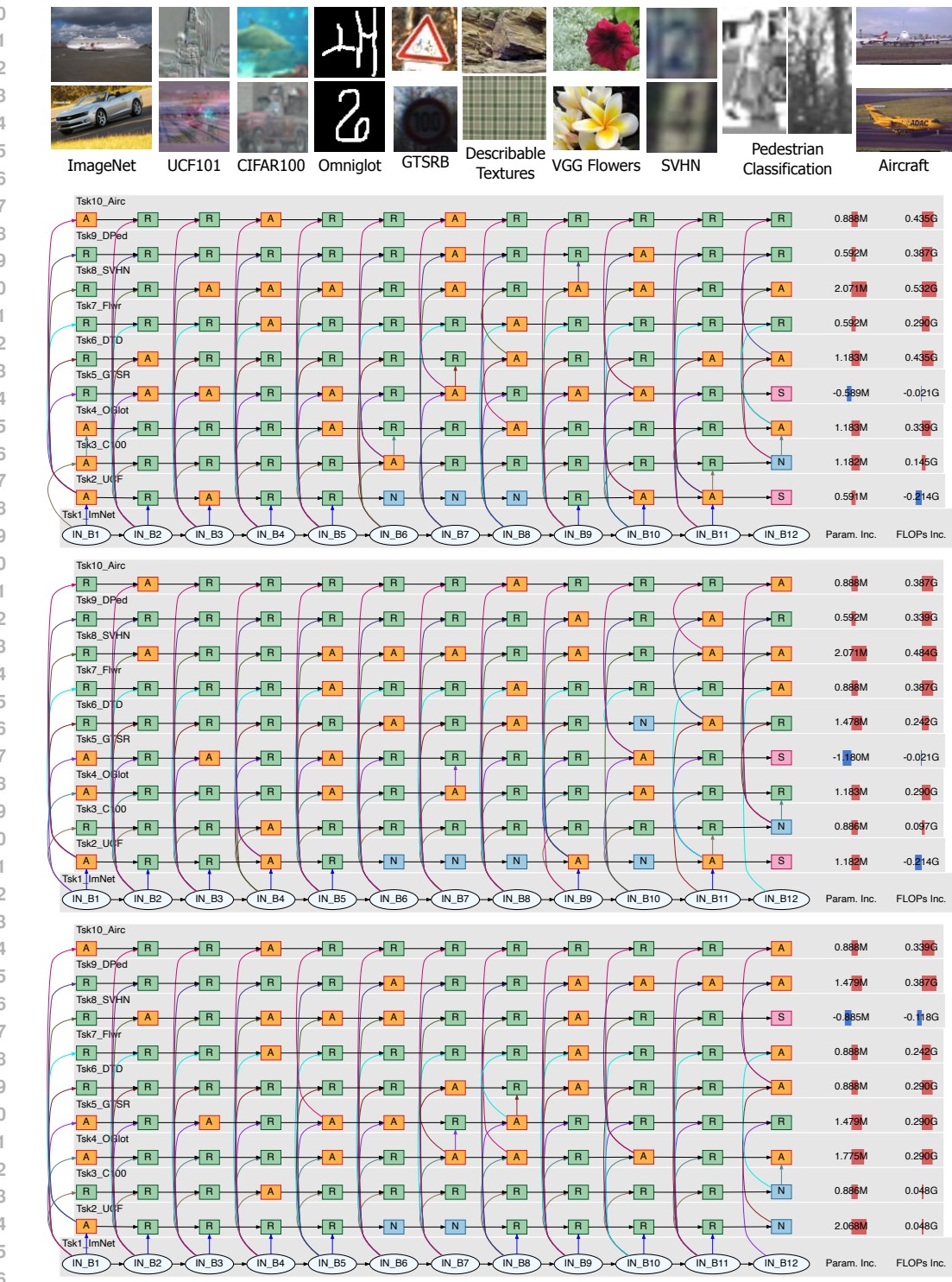

Figure 13: Examples of the task-synergy memory (CHEEM) learned on the VDD benchmark (Rebuffi et al., 2017a) with the task sequence shown in the top **using our proposed HEE-based NAS** and three different random seeds. The overall performance is reported in Table 9. S , R , A and N represent Skip, Reuse, Adapt and New respectively. The last two columns show the number of new task-specific parameters and added FLOPs respectively, in comparison with the first task, ImNet model. Overall, the learned task synergies make intutive sense and remain relatively stable across different random seeds.

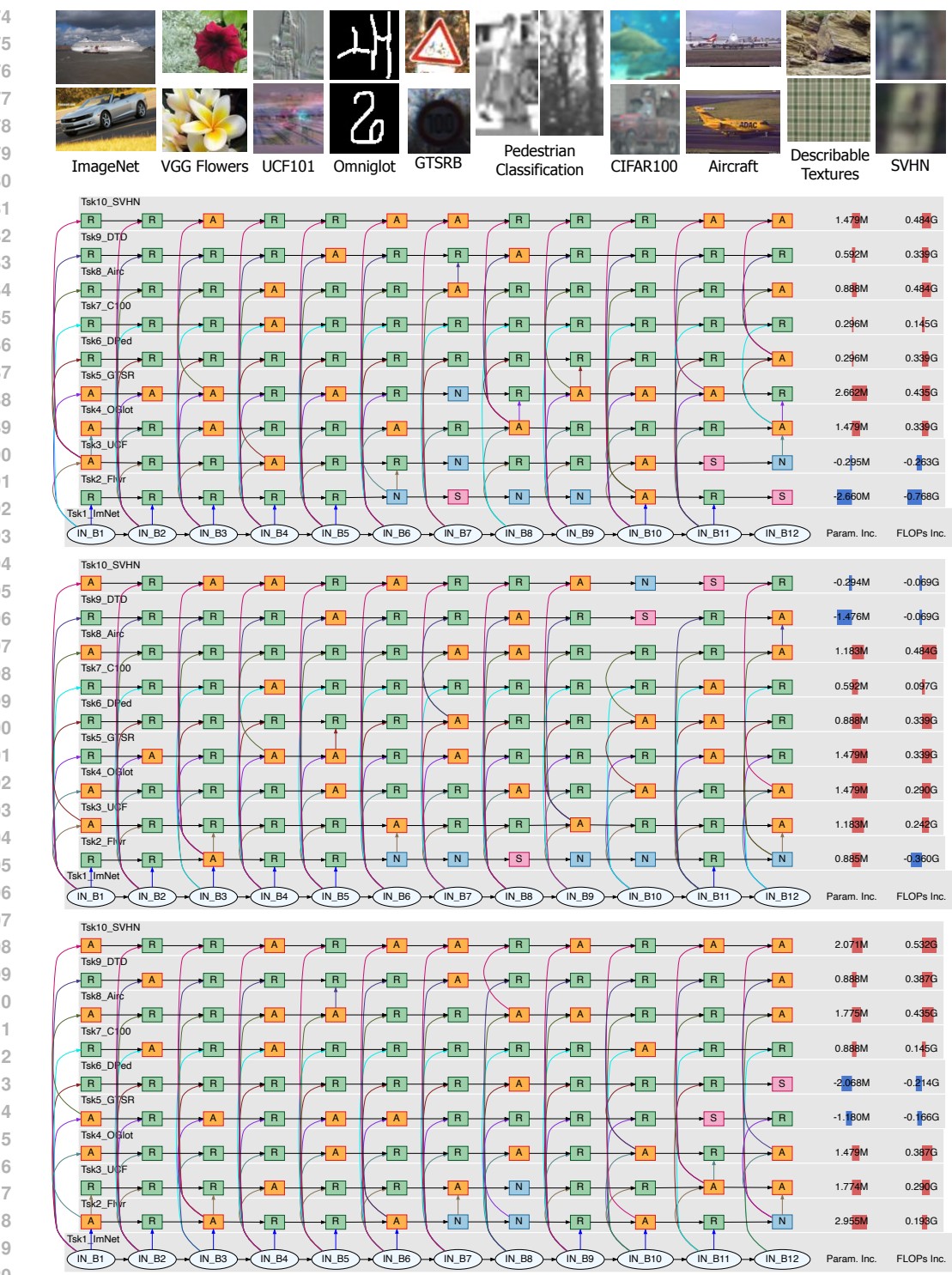

Figure 14: Examples of the task-synergy memory (CHEEM) learned on the VDD benchmark (Rebuffi et al., 2017a) with the task sequence shown in the top **using our proposed HEE-based NAS** and three different random seeds. The overall performance is reported in Table 9. S , R , A and N represent Skip, Reuse, Adapt and New respectively. The last two columns show the number of new task-specific parameters and added FLOPs respectively, in comparison with the first task, ImNet model. Overall, the learned task synergies make intuitive sense and remain relatively stable across different random seeds.

# E DATASET DETAILS

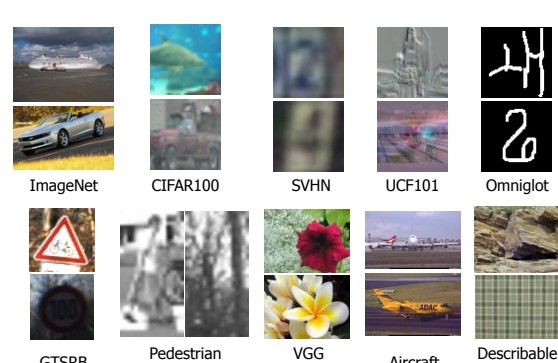

ImageNet  CIFAR100  SVHN  UCF101  Omniglot

GTSRB  Pedestrian Classification  VGG Flowers  Aircraft  Describable Textures

Figure 15: Example images from the VDD benchmark (Rebuffi et al., 2017a). Each task has a significantly different domain than others, making VDD a challenging benchmark for lifelong learning.

Table 10: The number of samples in training, validation and testing sets per task used in our experiments on the VDD benchmark (Rebuffi et al., 2017a).

| Task | Train | Validation | Test | #Categories |
|------|-------|------------|------|-------------|
| ImageNet12 | 1108951 | 123216 | 49000 | 1000 |
| CIFAR100 | 36000 | 4000 | 10000 | 10 |
| SVHN | 42496 | 4721 | 26040 | 10 |
| UCF | 6827 | 758 | 1952 | 101 |
| Omniglot | 16068 | 1785 | 6492 | 1623 |
| GTSR | 28231 | 3136 | 7842 | 43 |
| DPed | 21168 | 2352 | 5880 | 2 |
| VGG-Flowers | 918 | 102 | 1020 | 102 |
| Aircraft | 3001 | 333 | 3333 | 100 |
| DTD | 1692 | 188 | 1880 | 47 |

## E.1 THE VDD BENCHMARK

It consists of 10 tasks: ImageNet-1k (Russakovsky et al., 2015), CIFAR100 (Krizhevsky et al., 2009), SVHN (Netzer et al., 2011), UCF101 Dynamic Images (UCF) (Soomro et al., 2012; Bilen et al., 2016), Omniglot (Lake et al., 2015), German Traffic Signs (GTSR) (Stallkamp et al., 2012), Daimler Pedestrian Classification (DPed) (Munder & Gavrila, 2006), VGG Flowers (Nilsback & Zisserman, 2008), FGVC-Aircraft (Maji et al., 2013), and Describable Textures (DTD) (Cimpoi et al., 2014). All the images in the VDD benchmark have been scaled such that the shorter side is 72 pixels. Table 10 shows the number of samples in each task. Fig. 15 shows examples of images from each task of the VDD benchmark. In our experiments, we use 10% of `the official training data` from each of the tasks for validation (e.g., used in the target network selection in Section 3.2.3 in main text), and report the accuracy on `the official validation set` due to the unavailability of the ground-truth labels for the `official test data`. In Table 10, the `train, validation` and `test` splits are thus referred to 90% of the official training data, 10% of the official training data, and the entire official validation data respectively. When finetuning the learned architecture (i.e., the searched target network) for each task, we use `the entire official training data` to train and report results on `the official validation set`.

## E.2 IMAGENET-R BENCHMARK

The ImageNet-R(etention) (Hendrycks et al., 2021) dataset contains art, cartoons, deviantart, graffiti, embroidery, graphics, origami, paintings, patterns, plastic objects, plush objects, sculptures, sketches, tattoos, toys, and video game renditions of ImageNet classes. It has renditions of 200 ImageNet classes resulting in 30,000 images. The Split ImageNet-R benchmark proposed by (Wang et al., 2022c) uses the ImageNet-R dataset to build a continual learning benchmark specifically for studying methods that use model pretrained on ImageNet (Russakovsky et al., 2015). ImageNet-R poses challenges for such methods because of the diversity within the same class. The Split-ImageNet benchmark proves to be challenging for experience-replay based approaches because of this intra-class variance, as well as replay-free methods that use a frozen backbone from ImageNet as the accuracy of the standard models on ImageNet-R is low. We use the same training and validation splits as those used by (Smith et al., 2023a). For the balanced evaluation, we divide the data set into 10 tasks with 20 classes each, and report results across 3 runs with random class orders. For the imbalanced evaluation, we construct 6 tasks with 5, 10, 15, 20, 50 and 100 classes. We report results across 3 runs with tasks, with varying task orders.

# F  THE BASE VISION TRANSFORMER: VIT-B/8

We use the base Vision Transformer (ViT) model, with a patch size of $8 \times 8$ (ViT-B/8) model from (Dosovitskiy et al., 2021). The base ViT model contains 12 Transformer blocks. A Transformer block is defined by stacking a Multi-Head Self-Attention (MHSA) block and a Multi-Layer Perceptron (MLP) block with resudual connections for each block. ViT-B/8 uses 12 attention heads in each of the MHSA blocks, and a feature dimension of 768. The MLP block expands the dimension size to 3072 in the first layer and projects it back to 768 in the second layer. For all the experiments, we use an image size of $72 \times 72$ following the VDD setting. We base the implementation of the ViT on the `timm` package  (Wightman, 2019).

**Training the Base Model**   To train the ViT-B/8 model, we use the ImageNet data provided by the VDD benchmark (the `train` split in Table 10). To save the training time, we initialize the weights from the ViT-B/8 trained on the full resolution ImageNet dataset (224×224) and available in the `timm` package, and finetune it for 30 epochs on the downsized version of ImageNet (72×72) in the VDD benchmark. We use a batch size of 2048 split across 4 Nvidia Quadro RTX 8000 GPUs. We follow the standard training/finetuning recipes for ViT models. The file `cheem/artifacts/imagenet_pretraining/args.yaml` in our code folder provides all the training hyperparameters used for training the the ViT-B/8 model on ImageNet. During testing, we take a single center crop of 72×72 from an image scaled with the shortest side to scaled to 72 pixels.

# G  SETTINGS AND HYPERPARAMETERS IN LEARNING CHEEM

Starting with the ImageNet trained ViT-B/8, the proposed CHEEM learning consists of three components: *supernet training, evolutionary search for target network selection, and target network finetuning*.

Table 11: Data augmentations for the 9 tasks in the VDD benchmark.

| Task | Scale and Crop | Hor. Flip | Ver. Flip |
|---|---|---|---|
| CIFAR100 | Yes | p=0.5 | No |
| Aircraft | Yes | p=0.5 | No |
| DPed | Yes | p=0.5 | No |
| DTD | Yes | p=0.5 | p=0.5 |
| GTSR | Yes | p=0.5 | No |
| OGlt | Yes | No | No |
| SVHN | Yes | No | No |
| UCF101 | Yes | p=0.5 | No |
| Flwr. | Yes | p=0.5 | No |

Table 12: Data augmentations used for each task in the 5-Datasets benchmark.

| Task | Scale and Crop | Hor. Flip |
|---|---|---|
| MNIST | Yes | No |
| not-MNIST | Yes | No |
| SVHN | Yes | No |
| CIFAR100 | Yes | p=0.5 |
| Fashion MNIST | Yes | No |

**Data Augmentations**   A full list of data augmentations used for the VDD benchmark is provided in Table 11, and the data augmentations used for the tasks in the 5-datasets benchmark is provided in Table 12. The augmentations are chosen so as not to affect the nature of the data. Scale and Crop transformation scales the image randomly between 90% to 100% of the original resolution and takes a random crop with an aspect ratio sampled from a uniform distribution over the original aspect ratio $\pm 0.05$. In evaluating the supernet and the finetuned model on the validation set and test set respectively, images are simply resized to $72 \times 72$ with bicubic interpolation.

**Supernet Training**   *The VDD Benchmark*: For each task, we train the supernet for 100 epochs, unless otherwise stated. We use a label smoothing of 0.1. We use a learning rate of 0.001 and the Adam optimier (Kingma & Ba, 2015) with a Cosine Decay Rule. We use a batch size of 512, and ensure the minimum number of batches in an epoch is 15 (via repeatedly sampling when the number of total samples of a task is not sufficient). As stated in the paper, for the Exploration-Exploitation sampling scheme, we use an exploration probability $\epsilon = 0.3$.

*The 5-datasets Benchmark*: We use the same hyperparameters as those used in the VDD Benchmark, but train the supernet for 50 epochs to account for its relatively lower complexity.

*L2G with DARTS and $\beta$-DARTS*: We train the supernet of the Learn-to-Grow (L2G) (Li et al., 2019) for 50 epochs on the VDD benchmark and 25 epochs on the 5-datasets benchmark, since DARTS simultaneously trains all sub-networks (i.e. the entire supernet) at each epoch. We use a weight of 1 for the beta loss in all the experiments with $\beta$-DARTS.

**Evolutionary Search**  The evolutionary search is run for 20 epochs. We use a population size of 50. We use 25 candidates both in the mutation stage and the crossover stage. The top 50 candidates are retained. The crossover is performed among the top 10 candidates, and the top 10 candidates are mutated with a probability of 0.1. For the Exploration-Exploitation sampling scheme, we use an exploration probability $\epsilon = 0.5$ when generating the initial population.

**Finetuning**  The target network for a task selected by the evolutionary search is finetuned for 30 epochs with a learning rate of 0.001, Adam optimizer, and a Cosine Learning Rate scheduler. Drop Path of 0.25 and label smoothing of 0.1 are used for regularization. We use a batch size of 512, and a minimum of 30 batches are drawn.

We use a single Nvidia A100 GPU for all the experiments.

## H  MODIFYING SUPSUP, EFT AND LL TO WORK WITH VITS

In the main paper, we compare with Supermasks in Superposition (SupSup) (Wortsman et al., 2020), Efficient Feature Transformation (EFT) (Verma et al., 2021), and Lightweight Learner (LL) (Ge et al., 2023) in Table 5 under the task-to-task transfer learning paradigm. The three methods are originally developed for Convolutional Neural Networks. We modify them to be compatible with ViTs for a fair comparison with our CHEEM.

We use the same ViT-B/8 base model (Sec. F) for SupSup, EFT and LL. For the SupSup method (Wortsman et al., 2020), we learn masks for the weights of the final linear projection layer of the Multi-Head Self-Attention block using the straight through estimator (Bengio et al., 2013). We apply the EFT (Verma et al., 2021) on all the linear layers in the ViT-B/8 (i.e., all the Query/Key/Value projection layers, the final projection layer, and the FFN layers) by scaling their activation maps via the Hadamard product with learnable scaling vectors, following the original proposed formulation for fully-connected layers in the EFT (Verma et al., 2021). For the LL method (Ge et al., 2023) which learns a task-specific bias vector that is added to all the feature maps of convolutional layers, we learn a similar bias vector and add it to the output of all the linear layers of the ViT.

