# OpenReview forum: "Continual Learning via Learning a Continual Memory in Vision Transformer"
_ICLR.cc/2025/Conference — Submitted to ICLR 2025_

### Official Review · Reviewer_wv9n · 2024-11-01

**Soundness:** 3
**Presentation:** 1
**Contribution:** 2
**Rating:** 3
**Confidence:** 4

**Summary:**

This paper proposes a new model for task-incremental continual learning (TCL) by Vision Transformers. To improve the overall streaming-task performance of TCL, this work proposes a NAS-based method cooperated with a Hierarchical tasksynergy Exploration-Exploitation (HEE) sampling method, which effectively learns task synergies by structurally updating the identifed memory component. In this way, the proposed model can tackle various tasks in TCL by adaptively using reuse, adapt, new or skip strategies. The authors conduct experiments mainly on the challeging Visual Domain Decathlon benchmak, which demonstrates the effectiveness of the proposed model.

**Strengths:**

1. The work conducts experiments on the challenging Visual Domain Decathlon (VDD) benchmark, and obtains good results.
2. The work proposes a new model based on NAS and MoE, which is somewhat novel in continual learning field.

**Weaknesses:**

1. The writing and presentation of the paper is poor.
2. Why the authors ignore class-incremental learning in this work? Compared with task-incremental learning, class-incremental learning is more practical. Although the authors have explained some in Line 97-102, it is still confused to me.
3. The authors show an example of learned network architecture in Figure 4, but the conclusion is not clear to me. What insights can we get from the figure?
4. Please clarify the detailed differences between the adopted Visual Domain Decathlon (VDD) benchmark and other popular ones. More importantly, why the proposed model is specifically targeted at VDD? Please clarify this from the perspective of model design.
5. Please clarify why you choose the projection layer after MHSA. As you ilustrated in Line 199-202, please clarify this from the perspectives of forward transfer ability, forgetting, simplicity and invasive implementation.
6. Please conduct more comparison between your proposed model and existing models. For example, the authors only compare with S-Prompts and L2P in Table 2, it is not enough at all. Furthermore, I wonder if the authors have conducted a comparison with models at the same scale of parameters.

**Questions:**

See the Weakness.

---

> ### Author Response · Authors · 2024-11-24
> **Author Response (1/2)**
>
> Thank you for your valuable feedback. We address your comments as follows:
>
> ### Most important update in revision: **our proposed CHEEM works with Exemplar-free Class-Incremental Continual Learning (ExfCCL).** Please refer to our global response and revised manuscript for details.
>
> > Improvements in presentation
>
> We have revised the manuscript to improve the writing and presentation, and provided a brief summary of the changes in the global comment. Please refer to the revised manuscript at you convenience, as well as the global comment which summarizes the changes.
>
> > Addressing the class incremental setting
>
> We have modified the method to handle class-incremental learning. The modifications required to handle class-incremental learning are minor and do not change the core of the method. Section 2.1 and 2.2 in the revised manuscript describe the modifications in detail. We have reproduced them below for your convenience:
>
> "To handle class-incremental setting, we explicitely infer task IDs for test data, and adopt the method proposed in S-Prompts. For a task $t$, we leverage $K$-mean clustering of the CLS tokens of its training images computed by the query function $q(\cdot)$ (i.e., the base model $f_0$). Denote by $Z_t=\{z^1_t, \cdots, z^K_t\}$ the clustered task centroids for task $T_t$. Werefer to this as the external memory,  $\Psi=\cup_{t=1}^N Z_t$ after training (as opposed to the internal parameter memory). In inference, for a testing sample $x$, we first compute its {\tt CLS} token using the same query function $q(x)$, denoted by $z$. The task ID is then inferred via $K$-NN retrieval, $KNN(z, \Psi)$, either by retrieving the class ID of the Top-1 NN centroid in the external memory $\Psi$, or by majority voting of the class ID from the $K$-NN centroids."
>
> We have also added comparisons to Learn to Prompt, DualPrompt and CODAPrompt under the class incremental setting in Table 2 of the revised manuscript. The revised Table 2 gives a holistic overview of the advantages of CHEEM:
>
> - Comparing S-Prompts and CHEEM shows the advantage of using a dynamic backbone over a fixed one.
> - Learn to Prompt, DualPrompt and CODAPrompt show significant drops in accuracy due to discrepancy in the way the classification head is used in training and testing.
> - Baseline regularization-based methods EWC and simple L2 regularization show a strong baseline on VDD benchmark, even outperforming recent prompt-based method (due to the discrepancy in handling the classification head). However, they still suffer from significant catastrophic forgetting, highlighting the balanced stability and plasticity of CHEEM.
>
> > Purpose of Figure 4 (Figure 2 in the revised manuscript)
>
> Figure 4 in the original manuscript is meant to illustrate that the proposed HEE based NAS is effective in reusing the existing parameters (internal memory) when learning a new task. This is evidenced by the large number of Reuse and Adapt operations learned by the search. In contrast, the pure exploration sampling scheme learns a large number New operations adding to the parameter cost, and learns Skip operations in a manner that harms the overall performance (Figure 8 in the revised manuscript). This shows that the proposed hierarchical exploration-exploitation sampling is effective in learning a model that can reuse the existing parameters when learning a new task.

---

> ### Author Response · Authors · 2024-11-24
> **Author Response (2/2)**
>
> > Purpose of using the VDD benchmark over other benchmarks
>
> The primary motivation for utilizing the VDD benchmark is to address three critical aspects of continual learning:
>
> - The necessity for a natural benchmark in continual learning.
> - The need for a dynamic backbone to handle tasks that differ significantly from the base task.
> - The importance of explicit task ID inference in class-incremental learning.
>
> **Elaboration on Key Points**
>
> Previous benchmarks have often focused on constructing tasks within artificial settings. For instance, in the widely used Split ImageNet-R benchmark, tasks are typically created by randomly shuffling classes, with each task containing an equal number of classes. This design does not reflect real-world scenarios, where the number of classes within each task may vary significantly. The randomized classes in a task also do not reflect a natural setting: real world tasks can be coherent but still show a large variance in distribution. The VDD benchmark effectively addresses these limitations by providing a more realistic framework for evaluating continual learning models.
>
> _To illustrate this concept from a model design perspective_, we have included results on the Split ImageNet-R benchmark, a standard benchmark in prior research, within our revised manuscript. As shown in Table 3, methods such as Learn to Prompt, Dual Prompt, and CODAPrompt perform well under standard conditions, where each task contains an equal number of classes. However, their performance degrades substantially when the number of classes per task varies. This decline is primarily due to inconsistencies in how the classification head is utilized during training and testing. This effect is even more pronounced with the VDD benchmark, which exhibits a larger imbalance in class distribution across tasks.
>
> Under a class-incremental learning framework, Learn to Prompt, Dual Prompt, and CODAPrompt demonstrate significant accuracy drops on the VDD benchmark. Conversely, S-Prompts, which employs explicit task ID inference, performs considerably better. Building upon this strategy, we integrated a similar task ID inference approach into our proposed CHEEM model. Our findings show that CHEEM outperforms S-Prompts, underscoring the importance of employing a dynamic backbone to effectively handle tasks that differ substantially from the base task.
>
> > Choice of the Projection layer as CHEEM
>
> Our choice of the projection layer on the MHSA block is based on the following considerations:
>
> **Forward transfer ability**
> Fine-tuning the projection layer results in high average accuracy across all tasks in the VDD benchmark. While fine-tuning the entire MHSA block achieves slightly higher accuracy, the projection layer offers a more efficient, lightweight alternative.
>
> **Forgetting**
> Although fine-tuning the projection layer exhibits higher average forgetting compared to fine-tuning the Query and Key components, it significantly outperforms them in terms of forward transfer. Fine-tuning the entire MHSA block leads to significant forgetting, while the projection layer has a good balance between forward transfer and forgetting.
>
> **Simplicity and minimally invasive implementation**
> The Query, Key, and Value components are typically implemented as a single, monolithic linear layer in standard architectures. This structure limits the flexibility of these components to be modified with respect to the four basic operations proposed in our paper. In contrast, the projection layer is lightweight and a standalone component, making it easier to modify.
>
> > More comparisons with prior works
>
> We have added comparisons with DualPrompt and CODAPrompt in the revised manuscript (Table 2) under the class-incremental setting. Table 2 gives a holistic overview of the problem we address, as mentioneed above. We leave comparisons with models at the same scale of parameters for future work, as it is not the focus of this paper.

---

> > ### Author Response · Authors · 2024-11-26
> > **Request your feedback on our rebuttal**
> >
> > Dear Reviewer wv9n,
> >
> > Thank you for your valuable feedback again.
> >
> > We have carefully and significantly revised the submission based on all comments.
> >
> > Please check if we have addressed your concerns at your convenience.
> >
> > Thank you.

---

> > > ### Comment · Reviewer_wv9n · 2024-11-29
> > >
> > > After carefully reading other reviewers' comments and authors' response, I think the authors don't address my concerns at all.
> > > 1. About the writing: The authors responded in a really strange way. I expect the authors to polish the writing, .e.g, fix typos. However, the authors *almost re-write the manuscript*, in my view.
> > > 2. About the class-incremental setting: I expect the authors to explain the illustration in Line 97-102 in the original manuscript. However, the authors modify their method to fit class-incremental settings. It is a very significant change since the two settings are different.
> > > 3. About the benchmark and model design: I am still confused about why the proposed method is specific to the proposed benchmark. The better performance cannot explain why.
> > > 4. About more comparison: I think more comparison is needed. The authors only conduct a comparison with 4-5 previous methods. It is not convincing. And I would like to see more results on other traditional benchmarks, which is also mentioned by Reviewer 874z.
> > >
> > > Although I appreciate the authors' effort, I think this manuscript still has room for improvement. I suggest the authors to submit their manuscript to next conferences.

---

> ### Author Response · Authors · 2024-11-30
> **Thank you for your reply to our rebuttal**
>
> Dear Reviewer wv9n,
>
> Thank you for your reply.  We are sorry to hear that our rebuttal did not address your concerns.
>
> We would like to address your concerns in the reply as follows.
>
> > About the writing: The authors responded in a really strange way. I expect the authors to polish the writing, .e.g, fix typos. However, the authors almost re-write the manuscript, in my view.
>
> >> We significantly revised introduction and experiments due to the switch from task-incremental settings to class-incremental settings.  For the introduction, we revised it paragraph by paragraph to better reflect the update. For experiments, we re-organized and rewrote the experiments.  However, we keep the core component (section 2) largely the same, except for some rewording and a new subsection on task ID inference.  By analogy, we keep the "shape" of the submission, while significantly updating its "appearance".  The underlying reason is that we aimed to seize the opportunity of this review process (all valuable feedback) to try our best to make the submission better.
>
> > About the class-incremental setting: I expect the authors to explain the illustration in Line 97-102 in the original manuscript. However, the authors modify their method to fit class-incremental settings. It is a very significant change since the two settings are different.
>
> >> We agree that the changes are significant. But those changes are still based on the exactly same ideas proposed in the original submission, with a newly adopted task ID inference module.  More specifically, we did not need to retrain our models, and only change the inference settings by replacing given task IDs to inferred ones, and then rerun the evaluation scripts to compute the average accuracy and forgetting.  Motivated by this review process,  after we observed that we can indeed integrate task ID inference (adopted from S-Prompts) and our original dynamic backbone learning method,  we see this as a big opportunity since it reveals very interesting observations under the CCL settings. E.g., the state-of-the-art prompting-based methods (L2P++, DualPrompts and CODA-Prompt) catastrophically failed on VDD.
>
> > About the benchmark and model design: I am still confused about why the proposed method is specific to the proposed benchmark. The better performance cannot explain why.
>
> >> Our choice of sticking to the VDD benchmark is two-fold as follows: (i) It is the benchmark we claimed contributions in our original submission, so we want to keep it. (ii) In the literature of continual learning, after the Learn-to-Grow paper (Li, et al, ICML'19), we do not see other continual learning methods have tested this challenging benchmark. Since our method is motivated by Learn-to-Grow, we chose to focus on VDD. We observed very interesting results in our revision, e.g. the state-of-the-art prompting-based methods (L2P++, DualPrompts and CODA-Prompt) catastrophically failed on VDD. We also added ImageNet-R in the revision.
>
> >> In terms of "The better performance cannot explain why",  **we think we have figured out why.**  We added the analyses in the appendix of the revision which explained the experimental observations (why our method works well on VDD and why it does not work well on ImageNet-R, similarly for other prompting based methods).
>
> >> **Appendix Section A: Analysis of Task-Specific Head selection vs. Shared head.** We analyze the behavior of the final logits in prompt-based methods, which use a shared classification head, and investigate under which conditions the predicted class remains unchanged when the full head is included as opposed to when only the task-specific head segment is used.
>
> >> **Appendix Section B: Pros and Cons of task ID inference using centroids.**  We plot tSNE visualizations of the task centroids on both, the VDD and Split ImageNet-R benchmarks. We find that the centroids are well separated in the VDD benchmark, but not in the Split ImageNet-R benchmark. This is the reason for the lower performance of CHEEM in the balanced Split ImageNet-R setting compared to prompt-based methods.
>
> > About more comparison: I think more comparison is needed.
>
> >> We agree with you that evaluating our method on more datasets and comparing with more traditional methods will be helpful. We request you to re-think if our current comparisons are meaningful and/or interesting to the community or not. Comparing with more traditional methods in a fair manner will need to adapt them to work with ViTs, for which we have done for three methods, Supermasks in Superposition (SupSup) (Wortsman et al., 2020), Efficient Feature Transformation (EFT) (Verma et al., 2021) and Lightweight Learner (LL) (Ge et al., 2023) in the appendix (Table 6). Comparing on more benchmarks will take more time for experiments beyond what we have during the rebuttal period, we will try our best to add more in another round of revision.

---

### Official Review · Reviewer_874z · 2024-11-01

**Soundness:** 3
**Presentation:** 2
**Contribution:** 2
**Rating:** 3
**Confidence:** 5

**Summary:**

This paper presents a novel way of designing a dynamic network in continual learning.
Specifically, this work first analyzes the potential component of the ViT network as the synergy memory and concludes with choosing the projection layer.
The paper designs a NAS-based framework and defines operations like Reuse, New, Adapt, and Skip and associated methods to achieve continual learning.
The author also shows experiments under task-incremental learning settings on multiple datasets.

**Strengths:**

1. Strong Motivation. The natural way to conduct continual learning is to re-wire the neural networks and it is good to see the trial on the popular vision transformers.
2. Reasonable designing. I think the design of Reuse, New, Adapt, and Skip is reasonable and complete.
3. The proposed task-synergy exploration-exploitation (HEE) is novel and effective.

**Weaknesses:**

1. Limited applicability. This paper only focuses on the task-incremental learning paradigm, which holds a strong assumption that the incoming streaming samples are organized as different domain groups and that the task IDs are available during inference. I have noticed the authors have explained their thoughts on this in Lines 92-102. However, I am still concerned about this because (1) I don't think assuming a common classification space is a problem since identifying the samples within a domain is much more difficult than identifying which domain is a sample in. In other words, a common space is compatible with separated spaces, i.e., the unified classification space can be easily decomposed into domain-separated spaces but it is hard to merge independent spaces into a unified one. (2) The design of CHEEM is based on the task-incremental learning setting and I cannot see the potential of applying CHEEM in more complex continual learning settings (class-incremental, domain-incremental, online continual learning, etc.).

2. The design and the training of the proposed method CHEEM. The training of the proposed CHEEM relies on the feature mean of each layer's output. My concern is why using the mean of the features is enough to identify the similarities of different tasks but ignores other information like the (co-)variances of the features. I am very interested in how to measure the similarity of various tasks but this paper has not discussed more potential options of this.

4. Experiment results. (1) Why does the proposed method achieve worse performance on DAD shown in Table 4? I don't recognize the performance shown in Table 5. as SOTA performance since CHEEM loses on almost all datasets except the small dataset Aircraft. I think the authors did not make a reasonable explanation of the advantages of the proposed methods in the part of the experiment.
(2) It is stated in Lines 402 to 407 that the proposed method introduces fewer FLOPs compared to L2P. I am confused about how the number of `2.2G/task' in L2P. As far as I know, the L2P method maintains a fixed-size pool and only a sequential fixed-size prompt will be appended to the MHSA at all tasks so the increase of the FLOPs is fixed rather than increased per task. And the citation of L2P is not correct in the lines.
(3) Recent works on task-incremental learning have shifted into the vision-language models (like CLIP, which also employs the ViT as the backbone) [A, B], and I encourage the authors to explore the application of the proposed method in this challenging task.
```
    [A] Z. Zheng, M. Ma, K. Wang, Z. Qin, X. Yue, and Y. You. Preventing zero-shot transfer degradation in continual learning of vision-language
models. Proceedings of the IEEE/CVF Conference on Computer Vision and Pattern Recognition. 2023
    [B] Yu J, Zhuge Y, Zhang L, et al. Boosting continual learning of vision-language models via mixture-of-experts adapters. Proceedings of the IEEE/CVF Conference on Computer Vision and Pattern Recognition. 2024: 23219-23230.
```
5. The paper is hard to follow and the writing can be greatly improved, especially the structure of section 2.2.

6. Lack of citation of more vision-transformer-based continual learning methods:

```
[1] Jung D, Han D, Bang J, et al. Generating instance-level prompts for rehearsal-free continual learning. Proceedings of the IEEE/CVF International Conference on Computer Vision. 2023: 11847-11857.
[2] Tang Y M, Peng Y X, Zheng W S. When prompt-based incremental learning does not meet strong pretraining. Proceedings of the IEEE/CVF International Conference on Computer Vision. 2023: 1706-1716.
[3] Zhou D W, Sun H L, Ye H J, et al. Expandable subspace ensemble for pre-trained model-based class-incremental learning. Proceedings of the IEEE/CVF Conference on Computer Vision and Pattern Recognition. 2024: 23554-23564.
[4]Zhou D W, Ye H J, Zhang D C, Liu z. Revisiting Class-Incremental Learning with Pre-Trained Models: Generalizability and Adaptivity are All You Need, arXiv 2023.
```
And related NAS-based and archi-search-based methods:
```
[5]Gao Q, Luo Z, Klabjan D, et al. Efficient architecture search for continual learning. IEEE Transactions on Neural Networks and Learning Systems, 2022, 34(11): 8555-8565.
[6]Abati D, Tomczak J, Blankevoort T, et al. Conditional channel gated networks for task-aware continual learning. Proceedings of the IEEE/CVF conference on computer vision and pattern recognition. 2020: 3931-3940.
[7]Rajasegaran J, Hayat M, Khan S H, et al. Random path selection for continual learning. Advances in neural information processing systems, 2019, 32.
```

**Questions:**

Please see the weakness above.

---

> ### Author Response · Authors · 2024-11-24
> **Author Response (1/2)**
>
> Thank you for your valuable feedback. Below, we provide clarifications on the points raised:
>
> ### Most important update in revision: **our proposed CHEEM works with Exemplar-free Class-Incremental Continual Learning (ExfCCL).** Please refer to our global response and revised manuscript for details.
>
> > Applicabiltiy to class incremental setting
>
> We have modified the method to handle class-incremental learning. The modifications required to handle class-incremental learning are minor and do not change the core of the method. Section 2.1 and 2.2 in the revised manuscript describe the modifications in detail. We have reproduced them below for your convenience:
>
> "To handle class-incremental setting, we explicitely infer task IDs for test data, and adopt the method proposed in S-Prompts. For a task $t$, we leverage $K$-mean clustering of the CLS tokens of its training images computed by the query function $q(\cdot)$ (i.e., the base model $f_0$). Denote by $Z_t=\{z^1_t, \cdots, z^K_t\}$ the clustered task centroids for task $T_t$. Werefer to this as the external memory,  $\Psi=\cup_{t=1}^N Z_t$ after training (as opposed to the internal parameter memory). In inference, for a testing sample $x$, we first compute its {\tt CLS} token using the same query function $q(x)$, denoted by $z$. The task ID is then inferred via $K$-NN retrieval, $KNN(z, \Psi)$, either by retrieving the class ID of the Top-1 NN centroid in the external memory $\Psi$, or by majority voting of the class ID from the $K$-NN centroids."
>
> We have added comparisons with Learn to Prompt, DualPrompt, and CODAPrompt under the class-incremental learning setting, as presented in Table 2 of the revised manuscript. This updated table gives a comprehensive view of the strengths of CHEEM:
>
> - Advantage of a dynamic backbone: Comparing S-Prompts with CHEEM demonstrates the benefits of utilizing a dynamic backbone rather than a fixed one, leading to improved adaptability across tasks.
> - Methods such as Learn to Prompt, DualPrompt, and CODAPrompt exhibit significant accuracy drops due to inconsistencies in how the classification head behaves during training versus testing.  This shows the importance of explicit task ID inference.
> - Elastic Weight Consolidation (EWC) and simple L2 regularization serve as strong baselines on the VDD benchmark, outperforming recent prompt-based methods. This may be due to their more consistent handling of the classification head. However, these regularization methods still experience notable catastrophic forgetting, underscoring the balanced stability and plasticity achieved by CHEEM.
>
> We have also included experiments with the Split ImageNet-R benchmark (Section 3.2). We experiment with two settings: balanced setting, where ImageNet-R is divided into 10 tasks with 20 classes each, and an imbalanced setting, where ImageNet-R is divided into 6 tasks with uneven number of classes.
> - Under task-incremental setting, CHEEM outperforms all the baselines in both settings, showing the expressive power of the dynamic backbone.
> - Under class-incremental setting, prior prompt-based methods perform better than CHEEM in the balanced setting, which is caused due to the low average precision of the iask ID inference
> - In the imbalanced setting, prompt-based methods show a sharp drop in performance (e.g., from 76.48% to 67.66% by CODA-Prompt) due to the classification head discrepancy described above. Under the imbalanced setting, CHEEMperforms on par with the prompt-based methods.
>
> > Applicability to domain-incremental setting and online continual learning
>
> The VDD benchmark can be considered as a domain-incremental setting, the only difference being that the space of the outputs is also different. Hence, CHEEM can easily be extended to a domain-incremental setting if a separate classifier head is used for each domain. Moreover, the explicit task ID inference in the updated version of CHEEM can infer the domain, as shown by the performance on the VDD benchmark. We agree that the applicability of CHEEM to online continual learning is more difficult, and we plan to address this in future work. We also note that prior works using the ViT backbone have not addressed online continual learning, and as such it remains an open problem.

---

> ### Author Response · Authors · 2024-11-24
> **Author Response (2/2)**
>
> > Lines 92-102: Common classification space
>
> The meaning of lines 97 and 98 in the original manuscript was to simply state that current approaches assume the same number of classes per task and use a shared classification head, which is not a natural setting. We apologise for the confusion and we have carefully revised the entire section in the revised manuscript. Lines 96 to 130 in the revised manuscript state our motivation more clearly.
>
> We still argue that using a common classification head, where head segments from the previous tasks are masked out when learning the current task leads to performance degradation due to the discrepancy in the way the classification head is used in training and testing. This setup is used in most of the prior work like L2P, DualPrompt and CODA-Prompt. The drop is especially severe when the number of classes in a task varies, as shown in Table 3 in the revised manuscript.
>
> > Use of co-variances of features to guide the search
>
> This is an interesting question and worth future investigation. For the purpose of this work, we found that using the mean is sufficient to guide the search. We plan to address your point in future work.
>
> > Experimental results of Table 5 (Table 6 in the revised manuscript)
>
> We have clarified the purpose of these experiments in the Appendix Section D.3 of the revised manuscript. We reproduce the explanation below for your convenience:
>
> "Table 6 demonstrates that CHEEM achieves the highest average accuracy on the VDD benchmark. Although its performance on individual tasks may be lower, CHEEM exhibits robustness to domain variations, as reflected in its overall average accuracy. In contrast, other methods show varying performance depending on the domain. SupSup performs well on tasks that are significantly different from the base ImageNet task, such as UCF 101, Omniglot, and SVHN, but performs poorly on tasks closely related to ImageNet. Conversely, EFT and LL excel on tasks similar to ImageNet but struggle with out-of-distribution tasks. Despite its lower individual task performance, CHEEM's consistent performance across diverse downstream tasks highlights its adaptability and makes it more general."
>
> > FLOPs comparison with L2P
>
> The FLOPS increase in L2P is due to the additional prompt tokens added to the input sequence, which results in additional computations in the MHSA and FFN blocks. L2P adds a fixed number of prompts to every input sequence, and hence the increase in FLOPS is constant per task as you have stated. However, to avoid confusion, and since the increase is not significant enough to cause problems in pranctice, we have removed this in the revised version of the manuscript.
>
> > Application to Vision-Langiage Models
>
> Thank you for you suggestions and pointing out the potential of applying CHEEM to vision-language models. We will explore this in future work.
>
> > Clarity of writing
>
> We have revised the manuscript to improve the writing and presentation. Please refer to the revised manuscript at you convenience, along with the global coment highlighting the changes made.
>
> > Lack of citations
>
> Thank you for pointing out these excellent works to us. We have included citations to all the works mentioned.

---

> > ### Author Response · Authors · 2024-11-26
> > **Request your feedback on our rebuttal**
> >
> > Dear Reviewer 874z,
> >
> > Thank you for your valuable feedback again.
> >
> > We have carefully and significantly revised the submission based on all comments.
> >
> > Please check if we have addressed your concerns at your convenience.
> >
> > Thank you.

---

> ### Comment · Reviewer_874z · 2024-11-29
>
> Dear AC and authors of paper #8730:
>
> I have read the rebuttal and the revised version of the paper carefully and have further comments listed below.
>
> Yours,
> Reviewer 874z
>
> Comments:
>
> 1. I have noticed that the main claim of this work has undergone significant changes compared to both the original and revised versions. For example, in the original version (lines 103-104), it was stated that:
> “So, we choose to take one step forward by studying task-incremental learning to gain insights on how to structurally and dynamically update Transformer models at streaming tasks in the VDD benchmark.”
> However, in the revised version, the focus has shifted to addressing “the exemplar-free class-incremental (ExfCCL) setting.”
> In my original review, I suggested demonstrating the potential of the proposed method in a more challenging continual learning scenario rather than completely changing the main claim of the paper. Such a drastic change not only increases the reviewers’ burden but also causes confusion regarding the paper’s core contribution. Moreover, as per ICLR’s policy: “Area chairs and reviewers reserve the right to ignore changes that are significantly different from the original paper.”
>
> 2. Despite these issues, I have still carefully reviewed both the rebuttal and the revised paper. While I acknowledge the effort, I believe there is still room for improvement, and I do not recommend acceptance of the paper in its current state.
> Since the revised version now focuses on tackling exemplar-free class-incremental learning, I question the choice of sticking to the VDD benchmark.  Why not follow prior works[3,4] to conduct experiments on datasets CIFAR-100, CUB-200, ImageNet-R, ImageNet-A, Omni-Benchmark, Object, and VTAB? or follow the MTIL benchmark [A, B] to conduct experiments on Aircraft,  Caltech101, CIFAR100, EuorSAT, Flowers, Food, MNIST, OxfordPet, Cars and Sun397 sequentially? What makes the VDD dataset particularly unique or crucial for this study?
>
>     If the primary claim is now centered around class-incremental learning with ViTs, the experimental scope is not sufficiently extensive to convincingly validate the effectiveness of the proposed method.
> 3. The current approach leverages techniques from S-prompts for task ID inference. If class-incremental learning is now the main focus, more methods for task-ID inference should be explored and discussed. This is also related to my original review, where I raised concerns about how to measure task similarity—a point I believe the authors have not addressed thoroughly in either the rebuttal or the revised version.
>
> Overall, the experimental validation is not comprehensive enough, and critical discussions regarding task similarity and alternative methods are lacking.
>
> Based on the above points, I believe the current version of the paper is not ready for publication, and I have downgraded my score to 3.

---

> > ### Author Response · Authors · 2024-11-29
> > **Thank you for your feedback**
> >
> > Dear Reviewer 874z,
> >
> > Thank you for your feedback to our rebuttal and we are sorry to hear that we did not address your concerns.
> >
> > We would like to address the three points you raised in your feedback.
> >
> > > 1. In my original review, I suggested demonstrating the potential of the proposed method in a more challenging continual learning scenario rather than completely changing the main claim of the paper. Such a drastic change not only increases the reviewers’ burden but also causes confusion regarding the paper’s core contribution. Moreover, as per ICLR’s policy: “Area chairs and reviewers reserve the right to ignore changes that are significantly different from the original paper.”
> >
> > >> In our revision, the original core contribution (Hierarchical Exploration-Exploitation Sampling, HEE based NAS) is the same as the originally proposed method, with some rewording to be conciser. We did significantly revise the introduction and experimental results to reflect our changes of testing the proposed HEE-based sampling method.  **The switch from exemplar-free task-incremental settings in the original submission to the exemplar-free class-incremental settings in the revision is how we understood your original comment of "a more challenging continual learning scenario".**
> >
> > > 3. The current approach leverages techniques from S-prompts for task ID inference. If class-incremental learning is now the main focus, more methods for task-ID inference should be explored and discussed.
> >
> > >> We agree that it will be beneficial to test more methods for task-ID inference. However, it is not the focus and main contributions of our submission. So, we adopt the method proposed in S-Prompts for its simplicity, with which we did not need to retrain our models, and only change the inference settings by replacing given task IDs to inferred ones.  We think this is aligned with the ICLR policy to avoid too significant changes in our understanding.  We are sorry for causing any confusions.  We request you to re-think the potential of our proposed HEE-based method for continual learning.
> >
> > >  2. Despite these issues, I have still carefully reviewed both the rebuttal and the revised paper. ... I question the choice of sticking to the VDD benchmark. Why not follow ....
> >
> > >> We appreciate you for your big efforts going through our revision and rebuttal.
> >
> > >> We added ImageNet-R in the revision.  Our choice of sticking to the VDD benchmark is two-fold as follows: (i) It is the benchmark we claimed contributions in our original submission, so we want to keep it. (ii) In the literature of continual learning, after the Learn-to-Grow paper (Li, et al, ICML'19), we do not see other continual learning methods have tested this challenging benchmark. Since our method is motivated by Learn-to-Grow, we chose to focus on VDD.  We observed very interesting results in our revision, e.g. the state-of-the-art prompting-based methods (L2P++, DualPrompts and CODA-Prompt) catastrophically failed on VDD.
> >
> > >> Those being said, we agree with you that evaluating our method on more datasets and comparing with more traditional methods will be helpful.  We request you to re-think if our current comparisons are meaningful and/or interesting to the community or not.  Comparing with more traditional methods in a fair manner will need to adapt them to work with ViTs, for which we have done for three methods, Supermasks in Superposition (SupSup) (Wortsman et al., 2020), Efficient Feature Transformation (EFT) (Verma et al., 2021) and Lightweight Learner (LL) (Ge et al., 2023) in the appendix (Table 6). Comparing on more benchmarks will take more time for experiments beyond what we have during the rebuttal peroid, we will try our best to add more in another round of revision.
> >
> > > 3 how to measure task similarity—a point I believe the authors have not addressed thoroughly in either the rebuttal or the revised version.
> >
> > >> We added the analyses in the appendix of the revision which explained the experimental observations (why our method works well on VDD and why it does not work well on ImageNet-R, similarly for other prompting based methods).
> >
> > >>  **Appendix Section A: Analysis of Task-Specific Head selection vs. Shared head.** We analyze the behavior of the final logits in prompt-based methods, which use a shared classification head, and investigate under which conditions the predicted class remains unchanged when the full head is included as opposed to when only the task-specific head segment is used.
> >
> > >> **Appendix Section B: Pros and Cons of task ID inference using centroids** We plot tSNE visualizations of the task centroids on both, the VDD and Split ImageNet-R benchmarks. We find that the centroids are well separated in the VDD benchmark, but not in the Split ImageNet-R benchmark. This is the reason for the lower performance of CHEEM in the balanced Split ImageNet-R setting compared to prompt-based methods.

---

### Official Review · Reviewer_roRA · 2024-11-01

**Soundness:** 2
**Presentation:** 3
**Contribution:** 3
**Rating:** 8
**Confidence:** 4

**Summary:**

This paper focus on task-incremental continual learning (TCL). The proposed method, CHEEM, presents a hierarchical task-synergy exploration-exploitation sampling based NAS method for learning task-aware dynamic models continually with respect to four operations. The results on experiments conducted on the large-scale, diverse and imbalanced VDD benchmark demonstrate the better performance of CHEEM.

**Strengths:**

1. The proposed method proposes a novel hierarchical task-synergy exploration-exploitation sampling method based on NAS for learning task-aware dynamic models continually with respect to four operations.
2. This paper is good written and easy to follow.
3. The proposed method is evaluated on a persuasive experimental setup with larger datasets, which is similar with the wild scenarios.

**Weaknesses:**

1. There are not enough methods for comparison. In my opinion, the methodology of the paper may need to be compared with more methods，such as DualPrompt [1].

[1] DualPrompt: Complementary Prompting for Rehearsal-free Continual Learning. Wang. et.al. European Conference on Computer Vision.

**Questions:**

1. Due to the additional training time for NAS in this method, is the training time of the paper's method longer compared to other methods ?

---

> ### Author Response · Authors · 2024-11-24
> **Author Response**
>
> Thank your for your valuable feedback. Following are the clarifications on your comments:
>
> > More comparisons with prior works
>
> We have adapted the method to accommodate class-incremental learning. These modifications are minimal and do not alter the core framework of the original method. Section 2.1 and 2.2 in the revised manuscript describe the modifications in detail. We have reproduced them below for your convenience:
>
> "To handle class-incremental setting, we explicitely infer task IDs for test data, and adopt the method proposed in S-Prompts. For a task $t$, we leverage $K$-mean clustering of the CLS tokens of its training images computed by the query function $q(\cdot)$ (i.e., the base model $f_0$). Denote by $Z_t=\{z^1_t, \cdots, z^K_t\}$ the clustered task centroids for task $T_t$. Werefer to this as the external memory,  $\Psi=\cup_{t=1}^N Z_t$ after training (as opposed to the internal parameter memory). In inference, for a testing sample $x$, we first compute its {\tt CLS} token using the same query function $q(x)$, denoted by $z$. The task ID is then inferred via $K$-NN retrieval, $KNN(z, \Psi)$, either by retrieving the class ID of the Top-1 NN centroid in the external memory $\Psi$, or by majority voting of the class ID from the $K$-NN centroids."
>
> We have added comparisons with Learn to Prompt, DualPrompt, and CODAPrompt under the class-incremental learning setting, as presented in Table 2 of the revised manuscript. This updated table gives a comprehensive view of the strengths of CHEEM:
>
> - Advantage of a dynamic backbone: Comparing S-Prompts with CHEEM demonstrates the benefits of utilizing a dynamic backbone rather than a fixed one, leading to improved adaptability across tasks.
> - Methods such as Learn to Prompt, DualPrompt, and CODAPrompt exhibit significant accuracy drops due to inconsistencies in how the classification head behaves during training versus testing.
> - Elastic Weight Consolidation (EWC) and simple L2 regularization serve as strong baselines on the VDD benchmark, outperforming recent prompt-based methods. This may be due to their more consistent handling of the classification head. However, these regularization methods still experience notable catastrophic forgetting, underscoring the balanced stability and plasticity achieved by CHEEM.
>
> We have also included experiments with the Split ImageNet-R benchmark (Section 3.2). We experiment with two settings: balanced setting, where ImageNet-R is divided into 10 tasks with 20 classes each, and an imbalanced setting, where ImageNet-R is divided into 6 tasks with uneven number of classes.
> - Under task-incremental setting, CHEEM outperforms all the baselines in both settings, showing the expressive power of the dynamic backbone.
> - Under class-incremental setting, prior prompt-based methods perform better than CHEEM in the balanced setting, which is caused due to the low average precision of the iask ID inference
> - In the imbalanced setting, prompt-based methods show a sharp drop in performance (e.g., from 76.48% to 67.66% by CODA-Prompt) due to the classification head discrepancy described above. Under the imbalanced setting, CHEEMperforms on par with the prompt-based methods.
>
> With the new resutls and comparisons, we believe that the revised manuscript provides a more comprehensive comparison with prior works highlighting the advantages of CHEEM.
>
> > Additional training time for the proposed method
>
> The training time for the proposed method is longer compared to other methods, which is a drawback of the method. However, the proposed hierarchical exploration-exploitaiton sampling enables the method to explicitly reuse much of the parameters (internal memory) of the prior tasks, resulting in a small addition in the number of parameters. The search also enables learning the Skip operation, which reduces the number of FLOPs for a downstream task.

---

> > ### Comment · Reviewer_roRA · 2024-11-26
> >
> > The author has addressed my concerns, but considering the author's method is limited to task incremental learning, I still cannot improve my score

---

> > > ### Author Response · Authors · 2024-11-26
> > > **Thank you!  Our method works with Class-Incremental Learning Now!**
> > >
> > > Dear Reviewer roRA,
> > >
> > > Thank you.
> > >
> > > We have updated our method for Class-Incremental Learning with evaluations on both VDD and ImageNet-R.
> > >
> > > Could you please check our global comment and the revised submission at your convenience?
> > >
> > > Thank you very much.

---

> > > > ### Comment · Reviewer_roRA · 2024-11-26
> > > >
> > > > Thank you for addressing the feedback and updating your manuscript with evaluations on the VDD and ImageNet-R datasets. I appreciate the effort you have put into revising your work. I increase the score to 8.

---

> > > > > ### Author Response · Authors · 2024-11-26
> > > > > **Thank you!**
> > > > >
> > > > > Dear Reviewer roRA,
> > > > >
> > > > > Thank you very much. We appreciate your recognition of our work, and your valuable feedback that help us improving the work.
> > > > >
> > > > > Thanks.

---

### Official Review · Reviewer_ceyj · 2024-11-02

**Soundness:** 2
**Presentation:** 2
**Contribution:** 2
**Rating:** 5
**Confidence:** 4

**Summary:**

This work proposed to adapt a network for task-incremental learning. Considering the characteristics of different tasks, the proposed method learns additional parameters or reuse the model weight learned from previous tasks.
Experimental results demonstrate the effectiveness of the proposed method.
The presentation can be improvement.

**Strengths:**

1. Instead of simply learning more parameters, the proposed method considers reusing the model weight when the new task is similar with previous task.
2. Ablation study is conducted for each design component.

**Weaknesses:**

1. The presentation can be improve. For example, the figure 4 in the Introduction is overly detailed and contains redundant content. To illustrate the main idea, showing a case of only two or three sequential tasks are enough. For the Fig. 5, the corresponding description is far away from the figure, and a detailed figure for method will be helpful.

2. The overall designs are not focus to the continual learning. (i) For example, in line 244 "How to Adapt in a sustainable way" is related to the gradient vanishing or exploding when learning deep network. And the design of residual connection is not novel.
(ii) Using the cls-token to guide the task-specific learning is not a significant contribution because previous works also use the cls-token to select and learn task specific prompts as in L2P and DualPrompt. (iii) The proposed HEE-BASED NAS considers the task similarity but the novelty is limited due to (ii).

3. The methods compared in Table 2 and Table 3 are limited and somewhat outdated; the latest papers in class-incremental learning should be included with necessary modifications.

4. With the parameter-efficient finetuning (PEFT) techniques, the task-incremental learning seems can be directly addressed by learning different PEFT module while keeping the main backbone frozen. The L2P learns the tasks prompt, which can be regarded as prompt tuning in PEFT. Although simply adding more prompts will quickly leads to performance saturation, adding more PEFT modules such as Adapter, LoRA may help. Therefore, a natural question is the comparison of the proposed method and the combination of different PEFT compare in terms of performance and number of learnable parameters.

**Questions:**

Please refer to the Weakness.

---

> ### Author Response · Authors · 2024-11-24
> **Author Response (1/2)**
>
> Thank your for your valuable feedback. Following are the clarifications on your comments:
>
> ### Most important update in revision: **our proposed CHEEM works with Exemplar-free Class-Incremental Continual Learning (ExfCCL).** Please refer to our global response and revised manuscript for details.
>
>
> > Improvements to the presentation
>
> We have revised the manuscript to improve the presentation. We have simplified Figure 4 (Figure 2 in the revised manuscript) according to your suggestions. We have also made sure that Figure 5 in the original manuscript is closer to the corresponding description (Figure 3 in the revised manuscript).
>
> > The overall designs are not focus to the continual learning. (i) For example, in line 244 "How to Adapt in a sustainable way" is related to the gradient vanishing or exploding when learning deep network. And the design of residual connection is not novel. (ii) Using the cls-token to guide the task-specific learning is not a significant contribution because previous works also use the cls-token to select and learn task specific prompts as in L2P and DualPrompt. (iii) The proposed HEE-BASED NAS considers the task similarity but the novelty is limited due to (ii).
>
> The core focus of the paper is to formulate a method that enables dynamic allocation of parameters to tasks based on their difficulty, including the ability to skip modules when the task is easy. We have revised the manuscript to convey the motivations and contributions better in Section 1. We use the insights from prior works to enable us to implement our CHEEM framework, and our methodology explains how the different components are integrated effectively.
>
>
> > The methods compared in Table 2 and Table 3 are limited and somewhat outdated; the latest papers in class-incremental learning should be included with necessary modifications.
>
> We have adapted the method to accommodate class-incremental learning. These modifications are minimal and do not alter the core framework of the original method. Section 2.1 and 2.2 in the revised manuscript describe the modifications in detail. We have reproduced them below for your convenience:
>
> "To handle class-incremental setting, we explicitely infer task IDs for test data, and adopt the method proposed in S-Prompts. For a task $t$, we leverage $K$-mean clustering of the CLS tokens of its training images computed by the query function $q(\cdot)$ (i.e., the base model $f_0$). Denote by $Z_t=\{z^1_t, \cdots, z^K_t\}$ the clustered task centroids for task $T_t$. Werefer to this as the external memory,  $\Psi=\cup_{t=1}^N Z_t$ after training (as opposed to the internal parameter memory). In inference, for a testing sample $x$, we first compute its {\tt CLS} token using the same query function $q(x)$, denoted by $z$. The task ID is then inferred via $K$-NN retrieval, $KNN(z, \Psi)$, either by retrieving the class ID of the Top-1 NN centroid in the external memory $\Psi$, or by majority voting of the class ID from the $K$-NN centroids."
>
> We have added comparisons with Learn to Prompt, DualPrompt, and CODAPrompt under the class-incremental learning setting, as presented in Table 2 of the revised manuscript. This updated table gives a comprehensive view of the strengths of CHEEM:
>
> - Advantage of a dynamic backbone: Comparing S-Prompts with CHEEM demonstrates the benefits of utilizing a dynamic backbone rather than a fixed one, leading to improved adaptability across tasks.
> - Methods such as Learn to Prompt, DualPrompt, and CODAPrompt exhibit significant accuracy drops due to inconsistencies in how the classification head behaves during training versus testing.  This shows the importance of explicit task ID inference.
> - Elastic Weight Consolidation (EWC) and simple L2 regularization serve as strong baselines on the VDD benchmark, outperforming recent prompt-based methods. This may be due to their more consistent handling of the classification head. However, these regularization methods still experience notable catastrophic forgetting, underscoring the balanced stability and plasticity achieved by CHEEM.
> We have also included experiments with the Split ImageNet-R benchmark (Section 3.2). We experiment with two settings: balanced setting, where ImageNet-R is divided into 10 tasks with 20 classes each, and an imbalanced setting, where ImageNet-R is divided into 6 tasks with uneven number of classes.
> - Under task-incremental setting, CHEEM outperforms all the baselines in both settings, showing the expressive power of the dynamic backbone.
> - Under class-incremental setting, prior prompt-based methods perform better than CHEEM in the balanced setting, which is caused due to the low average precision of the iask ID inference
> - In the imbalanced setting, prompt-based methods show a sharp drop in performance (e.g., from 76.48% to 67.66% by CODA-Prompt) due to the classification head discrepancy described above. Under the imbalanced setting, CHEEMperforms on par with the prompt-based methods.

---

> ### Author Response · Authors · 2024-11-24
> **Author Response (2/2)**
>
> > Comparison with PEFT techniques
>
> The core focus of the paper is to formulate a method that enables dyynamic allocation of parameters to tasks based on their difficulty, including the ability to skip modules when the task is easy. A key drawback of PEFT methods used for continual learning is that they do not enable the model to dynamically allocate lesser compute to easier tasks. In the proposed CHEEM, the Skip operation enables this. Furthermore, the Adapt operation in the proposed method can be replaced by other adapter methods such as LoRA. We leave this study for future work, as it is not the focus of this paper.

---

> > ### Author Response · Authors · 2024-11-26
> > **Request your feedback on our rebuttal**
> >
> > Dear Reviewer ceyj,
> >
> > Thank you for your valuable feedback again.
> >
> > We have carefully and significantly revised the submission based on all comments.
> >
> > Please check if we have addressed your concerns at your convenience.
> >
> > Thank you.

---

### Author Response · Authors · 2024-11-24
**Brief summary of Results and Changes (1/2)**

We thank the reviewers for their valuable feedback. We have carefully revised the original manuscript based on the comments provided. Below, we briefly summarize our results, followed by an overview of the changes made in the revised manuscript. We believe that the revised manuscript is clearer and that the changes implemented in response to the reviewers' suggestions have significantly enhanced the overall quality of the paper. We have addressed the concerns raised by reviewers through individual comments.

## Summary of Changes
- As suggested by reviewers `ceyj`, `874z`, `wv9n`, we have revised the writing of the Introduction and Approach to improve clarity.
- Addressing the concerns raised by reviewers `874z` and `wv9n` about the applicability of CHEEM to class-incremental scenarios, we have modified the method to handle class-incremental learning. The modifications required to handle class-incremental learning are minor and do not change the core of the method.

### Section 1: Introduction
- The Introduction has been shortened and states the motivations and contributions more clearly.
- Following is the motivation for CHEEM, which is now clearly stated in the Introduction:

As shown in Figure 1, we formulate continual learning as a problem of learning continual memory in ViT, which has two components: (i) The internal memory enabling a dynamic learning-to-grow feature backbone that balances stability and plasticity, mitigating catastrophic forgetting through **task synergies, in which a new task learns automatically to  reuse/adapt modules from previous similar tasks, or to introduce new modules when needed, or to skip some modules when it appears to be an easier task**. The internal parameter memory learning presents alternative perspectives to the input and prefix prompting based methods. (ii) The external memory enabling task ID inference for test data, for which we adopt a method proposed in~\citep{s-prompts} for its simplicity.

**Core Contributions**
- We propose a hierarchical task-synergy exploration-exploitation sampling based NAS method for learning task-aware dynamic models continually with respect to four operations: Skip, Reuse, Adapt, and New to mitigate  catastrophic forgetting.
- We identify a "sweet spot" in ViT as the task-synergy internal (parameter) memory, i.e., the output projection layers after MHSA in ViT. It also presents a new usage for the class-token CLS in ViT as the internal memory updating guidance, in addition to leveraging it in maintaining the external (task-centroid) memory for task ID inference on the fly.
- This is the first work, to the best of our knowledge, to evaluate continual learning with ViTs on the large-scale, diverse and imbalanced VDD benchmark, with better performance than the prior art.

### Approach
- We have organized the contents to improve clarity.
- We have also added a detailed explanation of the method to handle class-incremental learning (Section 2.2).

### Experiments
We have added comparisons with Learn to Prompt, Dual Prompt, and CODAPrompt in a class-incremental setting (Table 2) and reorganised the Results to convey the intended message more clearly. We have also added experiments with the Split ImageNet-R benchmark, where we construct a task sequence with imbalanced number of classes in each task. Please see our response to reviewer `wv9n` for the reason for choosing the VDD benchmark for experiments.

Briefly, our results show that:

(i) A dynamic backbone is advantageous over a fixed one, as demonstrated by the comparison between S-Prompts and CHEEM.

(ii) Explicit Task ID inference is crucial for scenarios when the number of classes in a task varies, as shown by the comparison between Learn to Prompt, Dual Prompt, CODAPrompt, and CHEEM.

(iii) CHEEM enables a better balance between stability and plasticity compared to regularization based methods and experience replay.

(iv) Learn to Prompt, Dual Prompt, and CODAPrompt show significant drops in accuracy for the imbalanced split ImageNet-R (different number of classes in each task) compared to the balanced (same number of classes in each task) due to the discrepancy in the way the classification head is used in training and testing.

---

> ### Author Response · Authors · 2024-11-24
> **Brief summary of Results and Changes (2/2)**
>
> ### Appendix Section A: Analysis of Task-Specific Head selection vs. Shared head
> - We analyze the behavior of the final logits in prompt-based methods, which use a shared classification head, and investigate under which conditions the predicted class remains unchanged when the full head is included as opposed to when only the task-specific head segment is used.
>
> ### Appendix Section B: Pros and Cons of task ID inference using centroids
> - We plot tSNE visualizations of the task centroids on both, the VDD and Split ImageNet-R benchmarks. We find that the centroids are well separated in the VDD benchmark, but not in the Split ImageNet-R benchmark. This is the reason for the lower performance of CHEEM in the balanced Split ImageNet-R setting compared to prompt-based methods.

---

> ### Author Response · Authors · 2024-11-25
> **Summary of Updated Results 1/2**
>
> Dear reviewers and AC, thank you for you efforts in reviewing our submission. We have reproduced the updated results in Table 2 and Table 3 for your convenince. Please let us know if you have further questions.
>
> ## Results on VDD benchmark
>
> |Method  |  C100 |    SVHN |   UCF |  OGlt |  GTSR |  DPed |  Flwr |  Airc. |   DTD |  Avg. Acc | Avg. Forgetting | TCL Avg. Acc. | TCL Avg. Forgetting
> |-  |  - | - |   - |  - |  - |  - |  - |  - |  - | - | - | - | -
> L2P++ | 79.16 | 0.62 | 5.40 | 14.05 | 62.82 | 4.09 | 2.97 | 2.73 | 2.98 | 19.42 $\pm$ 0.09 | 5.49 $\pm$ 0.37 | 68.68 $\pm$ 0.67 | 10.25 $\pm$ 0.77
> DualPrompt  | 82.92 | 15.27 | 12.19 | 33.71 | 80.12 | 7.65 | 3.17 | 3.47 | 3.23 | 26.86 $\pm$ 0.40 | 3.54 $\pm$ 0.27 | 76.57 $\pm$ 0.23 | 2.57 $\pm$ 0.26
> CODA-Prompt  | 88.06 | 18.09 | 12.16 | 53.95 | 95.65 | 19.42 | 6.83 | 6.34 | 5.62 | 34.01 $\pm$ 0.99 | 8.0 $\pm$ 0.72 | 75.83 $\pm$ 0.60 | 6.25 $\pm$ 0.60
> S-Prompts  | 87.38 | 88.53 | 64.05 | 72.17 | 98.53 | **99.65** | **96.63** | 45.49 | **58.07** | 78.95 $\pm$ 0.07 | 0.32 $\pm$ 0.32 | 79.71 $\pm$ 0.08 | 0.0 $\pm$ 0.0
> L2 Regularization  | 82.59 | 65.57 | 51.57 | 26.83 | 95.62 | 98.50 | 83.04 | 33.80 | 51.67 | 65.47 $\pm$ 0.33 | 7.23 $\pm$ 0.51 | 69.63 $\pm$ 0.17 | 3.00 $\pm$ 0.47
> Experience Replay  | 56.68 | 6.23 | 33.45 | 74.33 | 7.00 | 0.02 | 64.25 | 27.03 | 37.55 | 34.06 $\pm$ 0.69 | 44.89 $\pm$ 0.72 | 54.55 $\pm$ 2.67 | 22.67 $\pm$ 3.01
> Our CHEEM | **88.56** | **95.63** | **75.05** | **83.81** | **99.15** | 99.64 | 90.92 | **55.53** | 56.42 | **82.74** $\pm$ 0.54 | 0.33 $\pm$ 0.00 | **84.65** $\pm$ 0.33 | 0.0 $\pm$ 0.0
>
> Some analyses of the results:
>
> **i) The Advantage of Task-Aware Dynamic Backbone Over Frozen Backbone:  CHEEM vs S-Prompts.** CHEEM adopts the same task ID inference method (i.e., the external memory) proposed by S-Prompts. The improvement by CHEEM with above 3\% absolute average accuracy increase clearly shows the advantage of our proposed internal parameter memory for maintaining task-aware dynamic backbones, against the method of prepending retrieved task-specific prompts in S-Prompts.
> The above Table  also shows a potential for harnessing prompt-based methods in CHEEM. S-Prompts performs better on tasks that are similar to the first ImageNet task and have less training data such as Flwr (918 training images) and DTD (1692 training images).
>
> **ii) Explicit Task ID Inference:  With vs Without.** The three baselines, L2P, DualPrompt and CODA-Prompt, use a shared head classifier in inference, which, as aforementioned in the introduction, suffers from the discrepancy between training and testing. Due to the significant imbalance class distributions in the VDD, their average accuracy undergo catastrophic drops (19.42\% by L2P, 26.86\% by DualPrompt and 34.01\% by CODA-Prompt, vs 78.95\% by S-Prompts and 82.74\% by our CHEEM).
>
> **iii) Stability vs Plasticity of the Backbone.** To further show the advantage of our CHEEM learning to update the feature backbone from the base model, we compare with EWC, L2 Regularization and Experience Replay. The three methods regularize the weights, including the shared head classifier, in learning new tasks using different formulations.  *Comparing with L2P, DualPromt and CODA-Prompt*, under CCL, the regularization based methods work better, showing the negative effects of the discrepancy of handling the shared head in training and testing by the three prompting based methods, as well as a potential of integrating them. *Comparing with S-Prompts*, the plasticity of updating the entire backbone with regularization is less effective than the plasticity introduced by prompts while keeping the backbone frozen in S-Prompts. *Comparing with our CHEEM*, the three regularization based methods suffer from significant forgetting, highlighting the balanced stability and plasticity via our CHEEM.

---

> ### Author Response · Authors · 2024-11-25
> **Summary of Updated Results 1/2 2/2**
>
> ## Results on Split ImageNet-R benchmark
>
> -|Imbalanced|-|-|- | Balanced|-|-|-
> |-| - | - | - |  -| -| -| -| -
> **Method**| **Avg. Acc** |  **Avg. Forgetting** | **TCL Avg. Acc**  |  **TCL Avg. Forgetting**  | **Avg. Acc** |  **Avg. Forgetting** | **TCL Avg. Acc**  |  **TCL Avg. Forgetting**
> L2P++ | 64.44 $\pm$ 1.38 | 8.62 $\pm$ 4.02 | 88.21 $\pm$ 0.28 | 0.50 $\pm$ 0.36 | 73.01 $\pm$ 0.57 | 5.80 $\pm$ 0.29 | 89.08 $\pm$ 0.38 |0.48 $\pm$ 0.02
> DualPrompt | 67.20 $\pm$ 1.38 | 6.95 $\pm$ 3.96 | 89.46 $\pm$ 0.57 | 0.45 $\pm$ 0.24 | 73.01 $\pm$ 0.55 | 3.35 $\pm$ 0.36 | 90.07 $\pm$ 0.19 | 0.29 $\pm$ 0.19
> CODAPrompt | **67.66** $\pm$ 3.09 | 7.06 $\pm$ 2.97 | 90.69 $\pm$ 0.44 | 0.46 $\pm$ 0.23 | **76.48** $\pm$ 0.25 | 5.04 $\pm$ 0.12 | 91.86 $\pm$ 0.34 | 0.45 $\pm$ 0.10
> Our CHEEM | 66.97 $\pm$ 0.43 | 8.44 $\pm$ 3.55 | **91.48** $\pm$ 0.63 | 0.0 $\pm$ 0.0 | 64.72 $\pm$ 0.28 | 8.73 $\pm$ 0.41 | **92.47** $\pm$ 0.64 | 0.0 $\pm$ 0.0
>
> **The ImageNet-R dataset is created to challenge models trained using ImageNet. So, by design, the external memory of our CHEEM that is based on ImageNet-1k trained base model will not work well.**  The above Table shows the results which we analyze as follows:
>
> **i) Randomly Assigned and \textit{Imbalanced} Classes in Streaming Task Challenge All of ExfCCL Methods We Test.**  Under CCL, the three prompting based methods suffer from catastrophic drop of performance from balanced to imbalanced settings (e.g., from 76.48 $\pm$ 0.25\% to 67.66 $\pm$ 3.09\% by CODA-Prompt, see Fig.5 and analyses in Appendix A ). Our CHEEM is on-par with DualPrompt and CODA-Prompt. Our CHEEM is also stable from balanced to imbalanced scenarios.  Under TCL, all methods obtain significant performance boost, and our CHEEM is better than all others. Similarly, we envision that  leveraging the dynamically learned backbone by our CHEEM in constructing the external memory will be a promising direction to be studied in future work.
>
> **ii) Randomly Assigned Yet *Balanced* Classes in Streaming Tasks Challenge the Task-Centroid based Task ID Inference.** On the one hand, for streaming tasks with  balanced but randomly sampled classes per task (without replacement), our CHEEM under CCL has the worst performance (64.72\% vs 76.48\% by CODA-Prompt), which was caused by the low average precision  of task ID inference (see Fig.6 and analyses in Appendix B). Our CHEEM under TCL obtains the best performance (92.47\%), which shows that the learned task-aware backbone is expressive. Overall, task centroids computed using the CLS tokens in the base model are not able to distinguish tasks from each other with high accuracy. One potential solution is to leverage the dynamically learned backbone by our CHEEM in constructing the external memory, considering its superior performance under TCL, at the expense of more costly task ID inference.  On the other hand, **although it is interesting to test continual learning approaches using the Split ImageNet-R benchmark, the random composition of classes in a task is not natural in comparison with scenarios in natural human learning. Unlike the Split ImageNet-R, the VDD benchmark may suit continual learning better from perspective the perspective of real-world scenarios, for which our CHEEM works the best**.

---

### Author Response · Authors · 2024-11-26
**Our proposed CHEEM works with Exemplar-free Class-Incremental Continual Learning (ExfCCL) in revision.**

Dear Reviewers,

Thank you for your efforts.  We have carefully and significantly revise the submission based on your valuable feedback.

> This paper explores continual learning (CL) using Vision Transformer (ViT) in streaming tasks under the challenging **exemplar-free class-incremental (ExfCCL) setting.**

>  We formulate ExfCCL as a learning problem consisting of two key sub-systems: **(i) task ID inference for test data**, which selects appropriate task-specific head classifiers to accounting for varying class distributions across tasks and streams, and **(ii) a dynamic learning-to-grow feature backbone that balances stability and plasticity, mitigating catastrophic forgetting through task synergies.**

>> To support task ID inference, we utilize an external memory mechanism that maintains task centroids computed by the base ViT throughout CL.

>> For the feature backbone, we identify optimal placements for internal (parameter) memory to enable a dynamic, task-synergy guided growing feature backbone. We propose **a Hierarchical Exploration-Exploitation (HEE) sampling-based neural architecture search (NAS)** method that effectively learns task synergies by continually and structurally updating internal memory with four basic operations: _reuse_, _adapt_, _new_, and _skip_.

> Our proposed CHEEM is evaluated on the challenging Visual Domain Decathlon (VDD) and ImageNet-R benchmarks, demonstrating its effectiveness.

Thank you.

---

### Author Response · Authors · 2024-11-30
**A summary of author-reviewer discussions and request reviewers' (re-)evaluation of the changes in revision**

Dear Reviewers,

We are grateful for your efforts and valuable time spent on our submission and revision, which have helped us a lot in improving the quality of our submission.

Reviewer **roRA** thought we have addressed her/his concerns in the rebuttal and revision. We appreciate the recognition.

We try to do our diligence to maximize the outcome from this valuable reviewing process.  So, we briefly summarize the current status as follows, and humbly request other reviewers' (re-)evaluation of the changes in our revision.

Reviewer **ceyj** has yet given feedback on our rebuttal. We hope we will have her/his valuable feedback soon, which we appreciate very much.

Reviewers **wv9n** and **874z**  have further concerns.  The main concerns and our rebuttal are as follows.

> Our revision has changed too much.

>>  Overall,  **the text has been revised significantly, while the core idea and contribution are kept the same as the original submission.**   By analogy, if we treat the two versions as two streaming "tasks" in the reviewing process,  the two versions are "similar tasks", rather than ``dissimilar tasks" due to sharing the same core idea.  We provided more explanations individually to Reviewers **wv9n** and **874z**.

> After re-evaluating our method using class-incremental continual learning (CCL) settings, more results and comparisons are needed.

>> What we have provided include:  results on VDD and ImageNet-R and the provided comparisons with the state-the-art prompting based methods (L2P++, DualPrompt, CODA-Prompt, S-Prompts),  traditional continual learning methods (EWC, L2 Regularization, Experience Replay), and three state-of-the-art methods under task-to-task transfer based settings (Supermasks in Superposition (SupSup) (Wortsman et al., 2020), Efficient Feature Transformation (EFT) (Verma et al., 2021) and Lightweight Learner (LL) (Ge et al., 2023) in the appendix (Table 6)).

>> We totally agree that more results and comparisons are helpful. What we request for re-evaluating our revision is whether the provided results, together with previous task-incremental results before revision, are sufficient to support the advantages of our proposed method.  We believe our results and analyses provide insights for continual learning and shed lights on future directions.  We respect reviewers' decisions, but would like to make sure we do our best explaining our efforts, as to maximize the outcome of this reviewing process.

Thank you very much.

---

### Meta-Review · Area_Chair_6ko7 · 2024-12-22

**Metareview:**

(a) Scientific Claims and Findings

The paper introduces a method for task-incremental continual learning using a network adaptation strategy. The proposed approach, CHEEM, employs a hierarchical task-synergy exploration-exploitation sampling based on neural architecture search (NAS) to learn task-aware dynamic models. The method focuses on reusing model weights for similar tasks and introduces operations like Reuse, New, Adapt, and Skip to achieve continual learning. Experimental results on the Visual Domain Decathlon (VDD) benchmark demonstrate the method's effectiveness, though the presentation and comparison with other methods could be improved.

(b) Strengths

Reviewer ceyj appreciates the method's consideration of reusing model weights for similar tasks and the inclusion of an ablation study. Reviewer roRA finds the hierarchical task-synergy exploration-exploitation sampling method novel and effective, with a well-written paper and a persuasive experimental setup. Reviewer 874z commends the strong motivation and reasonable design of the operations, as well as the novelty of the task-synergy exploration-exploitation method. Reviewer wv9n notes the good results obtained on the challenging VDD benchmark and the novelty of the NAS and MoE-based model.

(c) Weaknesses

Reviewer ceyj points out that the presentation needs improvement, with figures being overly detailed or poorly placed, and the design not being focused on continual learning. Reviewer roRA mentions the lack of comparison with more methods, such as DualPrompt. Reviewer 874z highlights the limited applicability to task-incremental learning, concerns about the training method, and the lack of convincing experimental results. Reviewer wv9n criticizes the poor writing and presentation, the exclusion of class-incremental learning, and insufficient comparisons with existing models.

(d) Decision Reasons

On balance, AC agrees with negative points raised by the reviewers which outweigh the positive points. Reviewer ceyj rates the paper as marginally below the acceptance threshold due to presentation issues and limited novelty. Reviewer roRA gives a higher rating, appreciating the novel method and good experimental setup. However, Reviewer 874z and Reviewer wv9n both recommend rejection, citing limited applicability, poor presentation, and insufficient comparisons. Importantly, after the rebuttal period, the most postitive reviewer  roRA, states the following
> I have carefully reviewed the authors' rebuttal, their revisions, and the feedback from other reviewers. My comments are as follows: ﻿ I commend the authors for their effort in addressing the concerns raised during the review process. I concur with reviewers wv9n and 874z that the substantial revisions required, particularly those involving the main claim of the paper, are beyond what is typically feasible or supported within the constraints of this review cycle. ﻿ Considering this, I believe the manuscript would benefit from being submitted to another venue. This would provide the authors with an opportunity to fully address these foundational concerns and present a more cohesive and well-supported narrative.

The overall consensus is that while the method shows potential, the weaknesses outweigh the strengths, leading to a decision to reject. While the method shows promise and novelty in its approach to task-incremental learning, the limited applicability, lack of comprehensive comparisons are significant drawbacks. The paper does not convincingly demonstrate superiority over existing methods, and the focus on task-incremental learning without addressing more practical scenarios like class-incremental learning limits its impact. Additionally, the writing and structure need improvement to enhance clarity and understanding. These weaknesses outweigh the strengths, leading to a decision to reject.

**Additional Comments On Reviewer Discussion:**

Summary of Discussion and Changes During the Rebuttal Period:

During the rebuttal period, the authors made significant revisions to the manuscript, including additional evaluations on the VDD and ImageNet-R datasets. Reviewer roRA appreciated these efforts and increased their score to 8, acknowledging the improvements made. However, Reviewer 874z expressed concerns about the substantial changes to the paper's main claim, which shifted focus from task-incremental learning to exemplar-free class-incremental learning. This change was seen as confusing and burdensome for reviewers, and the experimental scope was deemed insufficient to validate the new focus. Reviewer 874z downgraded their score to 3, citing a lack of comprehensive experimental validation and discussion on task similarity.
Reviewer wv9n also found the changes problematic, noting that the manuscript was almost rewritten, which was not expected. They were concerned about the shift to class-incremental settings and the lack of clarity on why the proposed method was specific to the VDD benchmark. The reviewer also felt that more comparisons with existing methods were needed. Reviewer wv9n suggested that the manuscript still required improvement and should be submitted to future conferences.
In the final discussion, Reviewer ceyj acknowledged the authors' efforts but agreed with the concerns raised by Reviewers wv9n and 874z about the substantial revisions and limited novelty of the proposed method. They highlighted the need for a better approach to measuring task similarity and difficulty, which was not adequately addressed. The most positive reviewer roRA also agreed that the revisions were beyond what could be supported in the current review cycle and suggested resubmission to another venue.

Final Decision:

The final decision of the reviewers leaned towards rejection. While Reviewer roRA initially increased their score, the substantial changes to the paper's main claim and the lack of comprehensive experimental validation led to a consensus that the manuscript was not ready for publication. The reviewers appreciated the authors' efforts but felt that the foundational concerns needed to be fully addressed in a more cohesive and well-supported narrative, which could be better achieved by submitting to another venue.

---

### Decision · Program_Chairs · 2025-01-22

Reject